# The role of piriform associative connections in odor categorization

Xiaojun Bao[1]*, Louise LG Raguet[2], Sydni M Cole[1], James D Howard[1], Jay A Gottfried[1,3]*

[1]Department of Neurology, Northwestern University Feinberg School of Medicine, Chicago, United States; [2]Department of Biology, École Normale Supérieure, Lyon, France; [3]Department of Psychology, Northwestern University Weinberg College of Arts and Sciences, Evanston, United States

**Abstract** Distributed neural activity patterns are widely proposed to underlie object identification and categorization in the brain. In the olfactory domain, pattern-based representations of odor objects are encoded in piriform cortex. This region receives both afferent and associative inputs, though their relative contributions to odor perception are poorly understood. Here, we combined a placebo-controlled pharmacological fMRI paradigm with multivariate pattern analyses to test the role of associative connections in sustaining olfactory categorical representations. Administration of baclofen, a GABA(B) agonist known to attenuate piriform associative inputs, interfered with within-category pattern separation in piriform cortex, and the magnitude of this drug-induced change predicted perceptual alterations in fine-odor discrimination performance. Comparatively, baclofen reduced pattern separation between odor categories in orbitofrontal cortex, and impeded within-category generalization in hippocampus. Our findings suggest that odor categorization is a dynamic process concurrently engaging stimulus discrimination and generalization at different stages of olfactory information processing, and highlight the importance of associative networks in maintaining categorical boundaries.

*For correspondence: xiaojunbao2011@u.northwestern. edu (XB); j-gottfried@ northwestern.edu (JAG)

**Competing interests:** The authors declare that no competing interests exist.

## Introduction

Object categorization is an adaptive function of the brain, allowing organisms to sort information from the external world into behaviorally relevant classes. Importantly, sensory systems must generalize across different objects sharing similar features, but at the same time maintain the specificity of individual objects and categories (*Roach, 1978*; *Riesenhuber and Poggio, 2000*). Mechanisms of pattern recognition have been proposed to underlie the neural basis of object categorization, which requires a balance between generalizing inputs across a certain range of variations (known as *pattern completion*) and discriminating between distinct inputs (known as *pattern separation*) (*Riesenhuber and Poggio, 2000*; *Haberly, 2001*; *Wilson and Sullivan, 2011*; *Chapuis and Wilson, 2012*). Such computations can be achieved by associating sensory inputs with internal templates that are established through a lifetime of experience and encoded into memory (*Bar, 2007*).

Most neuroscientific research on pattern recognition has concentrated on the visual system, where associative areas in the visual ventral stream and the CA3 region of the hippocampus have been shown to support processes of object categorization (*Riesenhuber and Poggio, 2000*; *Yassa and Stark, 2011*; *Haxby et al., 2001*). In the olfactory system, information in a whiff of scented air is transformed into distributed patterns of neural activity in the piriform cortex, with both animal and human studies demonstrating that different odor objects evoke distinguishable ensemble activity patterns without spatial topography (*Wilson and Sullivan, 2011*; *Gottfried, 2010*; *Bekkers and Suzuki, 2013*; *Stettler and Axel, 2009*; *Howard et al., 2009*). Recent work has

**eLife digest** Imagine bringing your groceries home and tucking them into the refrigerator. You'll probably organize the items by categories: lemons and oranges into the fruit drawer, carrots and cauliflower into the vegetable drawer. Categorization is essential, allowing us to interact with the world in the most efficient way possible. If the differences between objects are not relevant to the task at hand, the brain will group objects together based on their shared properties and develop mental representations of the "categories". Importantly, we are still aware of the distinctions between objects within the same category.

Categories of odor (for example, minty or fruity) are represented in a part of the brain called the olfactory (or piriform) cortex, which receives information from odor cues as well as "top-down" information from other areas of the brain. But how do these top-down pathways influence odor categorization?

Bao et al. asked how the brain solves the problem of categorizing odors. For the experiments, human volunteers smelled six familiar odors belonging to three different categories while their brain activity was monitored using a magnetic resonance imaging (fMRI) scanner. Then, half of the participants were given a drug called baclofen that prevents top-down inputs, but not odor cues, from reaching the piriform cortex, while the rest received a placebo. After five days of taking the medication, all of the volunteers had another session of fMRI where they had to categorize the same odors as before.

The experiments show that when comparing the fMRI scans before and after the drug treatment, the representations of odors belonging to the same category became more distinct in the piriform cortex in the placebo group. Put differently, as the volunteers were repeatedly exposed to odors of well-known categories, they became better at discriminating individual odors within the same category. However, these changes were disrupted in the group of volunteers that took baclofen.

Bao et al.'s findings indicate that this "practice makes perfect" approach to recognizing odors relies on top-down inputs into the piriform cortex. In future work it will be important to study the roles of these inputs in learning new categories of odors, and to investigate whether the mechanisms identified here apply to other sensory information and to more abstract knowledge.

revealed that fMRI multivariate patterns in posterior piriform cortex (PPC) encode not only odor identity, but also category information (e.g., minty or woody), whereby odor patterns belonging to the same category are more similar (more overlapping) than those across different categories (*Howard et al., 2009*). Despite these insights, the mechanisms by which olfactory inputs are organized into categorical percepts through their associations with olfactory cortical areas are poorly understood.

The neural architecture of the piriform cortex makes it an attractive model for investigating mechanisms of odor object recognition. As the largest subregion of primary olfactory cortex, the piriform cortex receives afferent (bottom-up) inputs from the olfactory bulb through the lateral olfactory tract, and extensive associative (top-down) inputs from higher-order association areas such as orbitofrontal cortex (OFC), amygdala, and entorhinal cortex (*Carmichael et al., 1994*; *Johnson et al., 2000*; *Haberly and Price, 1978*; *Insausti et al., 1987*; *Insausti et al., 2002*). This convergence of bottom-up and top-down projections, along with the presence of dense recurrent collaterals, is thought to support olfactory pattern recognition and associative learning (*Haberly, 2001*; *Haberly and Bower, 1989*; *Wilson, 2009*). For example, when confronted with highly overlapping odor mixtures, rats can learn to discriminate or ignore detectable differences between these mixtures, with piriform activity patterns exhibiting either separation (enhanced discrimination) or completion (enhanced generalization), respectively (*Chapuis and Wilson, 2012*). Evidence from humans has also pointed to PPC as a substrate for odor discrimination (*Li et al., 2008*) and categorization (*Howard et al., 2009*). Together these findings suggest that piriform cortex is capable of modulating pattern representations along a discrimination-generalization spectrum in order to encode behaviorally adaptive meaning through perceptual experience.

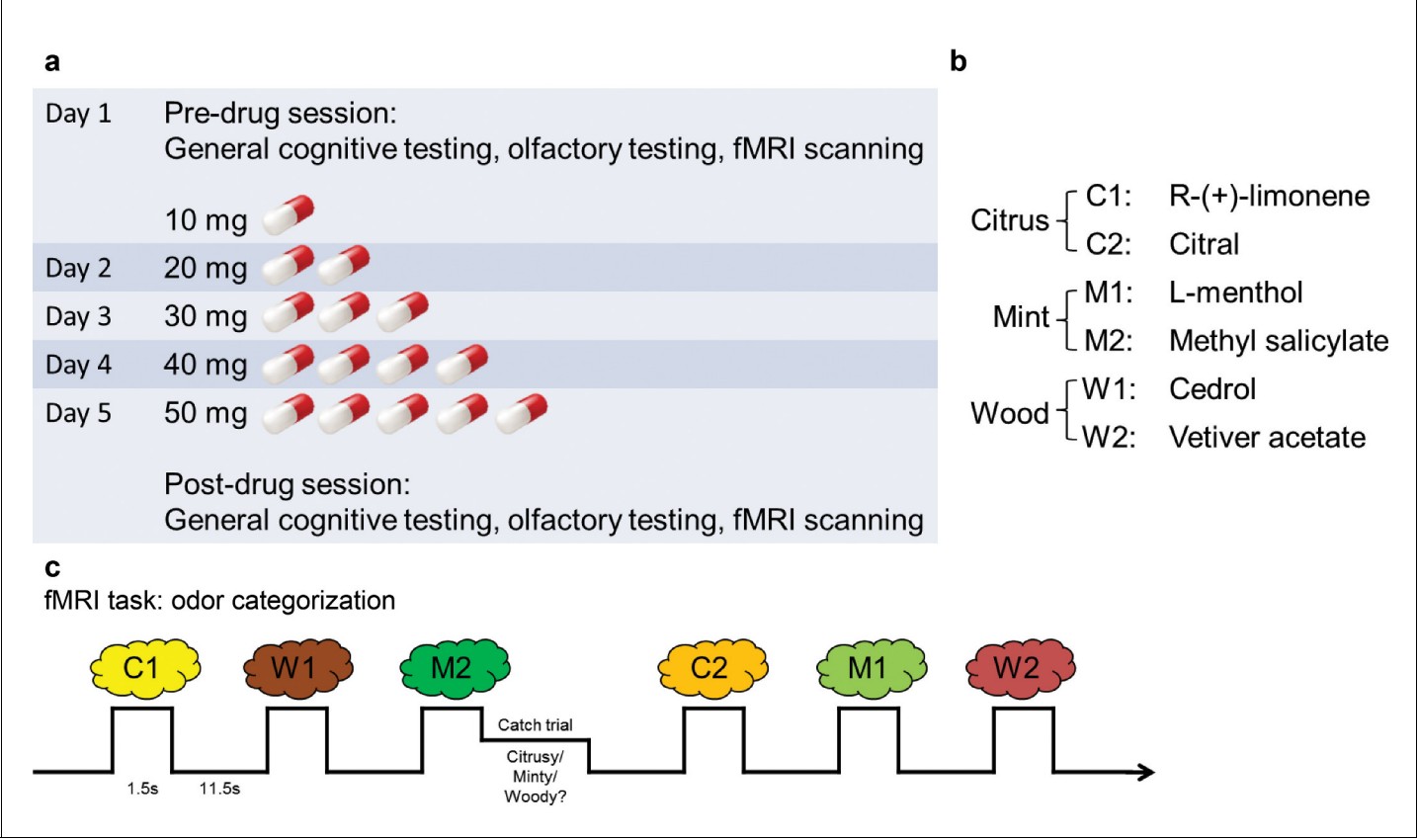

**Figure 1.** Experimental design. (a) Timeline of the 5-d experiment involving pre- and post-drug test sessions and the drug administration schedule. (b) The six odorants included two stimuli for each of the three categories (citrus, mint, and wood). (c) Paradigm of the fMRI odor categorization experiment. Subjects were prompted to sniff when an odorant was presented. They were asked to focus on the quality of the odor. In 14% of the trials (designated as catch trials), after the odor presentation, a screen with the names of the three categories appeared and subjects indicated the category of the received odor with a mouse click.

While theoretical modelling and empirical evidence propose that piriform associative connections are essential for odor recognition (*Haberly, 2001*), few studies have explicitly investigated the relative contributions of afferent inputs versus associative networks in supporting odor categorization. In a previous fMRI study, human subjects were deprived of afferent sensory input for one week, resulting in a reduction of odor-evoked mean activity in PPC, without alteration of pattern-based piriform representations of odor categories (*Wu et al., 2012*). Here we address the inverse question, namely, how attenuation of piriform associative connections influences odor category coding in primary sensory regions and higher-order cortical areas.

To this end, we took advantage of the GABA(B) receptor agonist, baclofen, to modify the relative balance between afferent and associative inputs within piriform cortex. Baclofen selectively suppresses synaptic transmission of association fibers into piriform cortex, but leaves afferent inputs from the olfactory bulb unaffected (*Tang and Hasselmo, 1994*). In vivo local application of baclofen in the piriform cortex of anesthetized rats modified the strength of odor-evoked responses of pyramidal neurons, by blocking broadly-tuned neurons and increasing odor-selective responses (*Poo and Isaacson, 2011*). In behaving animals, injection of baclofen into the piriform cortex following an olfactory fear conditioning session resulted in fear memory generalization, indicating that piriform associative connections are essential for consolidation of stimulus-specific memories (*Barnes and Wilson, 2014*).

Inspired by these animal studies, we conducted a double-blind, placebo-controlled drug study in human subjects to examine fMRI ensemble representations of familiar odor categories before and after treatment with baclofen. Given that odor object codes take the form of distributed ensemble

**Table 1.** Behavioral performance.

| Task | Placebo (n = 18) | | Baclofen (n = 14) | | P value of group × session interaction |
|---|---|---|---|---|---|
| | Pre | Post | Pre | Post | |
| MMSE | 29.89 ± 0.11 | 29.94 ± 0.06 | 29.93 ± 0.07 | 30.00 ± 0 | 0.86 |
| Digit span (forward) | 7.22 ± 0.22 | 7.72 ± 0.14 | 7.14 ± 0.31 | 7.43 ± 0.20 | 0.55 |
| Digit span (backward) | 6.00 ± 0.20 | 6.00 ± 0.29 | 5.57 ± 0.31 | 5.71 ± 0.27 | 0.69 |
| Trail making test B (s) | 48.32 ± 2.70 | 39.05 ± 2.01 | 57.62 ± 5.42 | 47.11 ± 5.67 | 0.85 |
| Stanford sleepiness scale | 2.44 ± 0.17 | 2.22 ± 0.21 | 1.93 ± 0.20 | 2.50 ± 0.33 | 0.041* |
| Sniffin' Sticks (odor detection threshold) | 7.08 ± 0.84 | 9.65 ± 1.06 | 7.82 ± 0.95 | 8.57 ± 1.01 | 0.22 |
| UPSIT (odor identification) | 36.28 ± 0.61 | 36.00 ± 0.56 | 34.57 ± 0.49 | 33.79 ± 0.63 | 0.55 |
| α- vs. β-pinene triangle test (fine odor discrimination) | 0.66 ± 0.05 | 0.72 ± 0.05 | 0.72 ± 0.05 | 0.73 ± 0.07 | 0.45 |
| Odor intensity ratings | 4.00 ± 0.32 | 4.13 ± 0.31 | 3.05 ± 0.18 | 2.91 ± 0.28 | 0.39 |
| Odor pleasantness ratings | 5.43 ± 0.16 | 5.63 ± 0.16 | 5.64 ± 0.13 | 5.63 ± 0.16 | 0.12 |
| Odor category descriptor ratings (within − across) | 7.47 ± 0.44 | 7.49 ± 0.36 | 7.44 ± 0.46 | 7.78 ± 0.34 | 0.59 |
| Odor pairwise similarity ratings (within − across) | 4.16 ± 0.60 | 5.14 ± 0.55 | 3.93 ± 0.32 | 4.53 ± 0.44 | 0.59 |
| Odor categorization catch trial accuracy | 0.87 ± 0.04 | 0.89 ± 0.03 | 0.81 ± 0.04 | 0.81 ± 0.04 | 0.80 |
| Odor categorization catch trial RT (s) | 3.29 ± 0.23 | 2.85 ± 0.15 | 3.89 ± 0.38 | 3.48 ± 0.34 | 0.93 |
| Visual categorization catch trial accuracy | 0.97 ± 0.01 (n = 14) | 0.99 ± 0.004 | 0.97 ± 0.01 (n = 11) | 0.96 ± 0.01 | 0.21 |
| Visual categorization catch trial RT (s) | 0.42 ± 0.02 | 0.40 ± 0.03 | 0.44 ± 0.04 | 0.52 ± 0.06 | 0.25 |

Data are shown for cognitive and olfactory tests, as well as for behavioral performance in fMRI experiments from placebo and baclofen groups in pre- and post-drug sessions. Scores are presented as mean ± s.e.m. P values reported are for the interaction effects between group and session, based on a 2-way ANOVA, with one between-group 'drug' factor (placebo/baclofen) and one within-subject 'session' factor (pre/post). *P < 0.05.

patterns, we used multivariate fMRI analyses to characterize baclofen effects in olfactory areas found to represent categorical information. The placebo group served as a control to account for session-effect confounds between pre- and post-drug phases of the study. As such, we examined the effects of baclofen by comparing pre-to-post changes relative to those observed in placebo subjects (i.e., group-by-session interaction). We predicted that baclofen would disrupt associative connections, leading to perceptual and neural reorganization of odor categories in piriform cortex and in olfactory downstream areas including OFC, amygdala, entorhinal cortex, and hippocampus.

## Results

The experiment spanned 5 days (*Figure 1a*). On day 1, subjects underwent pre-drug cognitive and psychophysical testing and fMRI scanning (*Figure 1c*). They were subsequently administered either placebo (n = 18) or increasing doses of baclofen (n = 14) for 5 consecutive days, in a double-blind design. This 5-day schedule was adopted to reach a target dose of 50-mg baclofen while minimizing the occurrence of side effects (*Terrier et al., 2011*). After taking the final dose on day 5, subjects underwent the same testing and fMRI scanning procedures as in the pre-drug session. During scanning, subjects completed an olfactory categorization task, as well as a control visual categorization task to establish the sensory specificity of the imaging findings.

### General cognition and olfactory perception

We first established that baclofen did not generally compromise cognitive or perceptual performance. Specifically, we found no significant differences between baclofen and placebo groups on neuropsychological assessments of basic cognition, short-term memory, visual attention, or task switching (*Table 1*). We also collected subjective reports of sleepiness using the Stanford Sleepiness Scale (SSS) during test sessions, given that the most common adverse reaction to baclofen

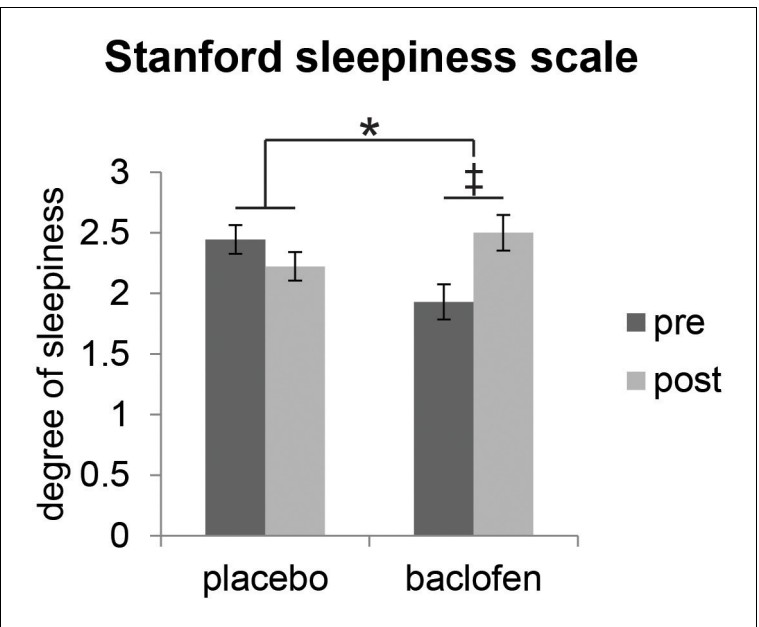

**Figure 2.** Effect of baclofen on subjective sleepiness. Ratings from the Stanford Sleepiness Scale (1 = 'wide awake', 7 = 'sleep onset soon', mean ± within-subject s.e.m., placebo n = 18, baclofen n = 14) indicate that there was a significant interaction between drug groups (placebo vs. baclofen) and session (pre vs. post) ($F_{1,30}$ = 4.57, P = 0.041; *P < 0.05). Post-hoc within-group comparisons showed no effect of session in placebo subjects ($F_{1,17}$ = 0.88, P = 0.36), and a marginal effect of session in baclofen subjects ($F_{1,13}$ = 3.85, ‡P = 0.072).

medication is transient drowsiness (*RxList The internet Drug Index, 2007*). Baclofen subjects reported feeling sleepier after taking the drug (*Figure 2*), though reaction times during the fMRI categorization task did not differ from placebo subjects (*Table 1*). Finally, we examined whether baclofen altered general odor perception. Placebo and baclofen groups did not differ on olfactory measures of detection threshold, identification, fine odor discrimination (*Figure 3d*), or intensity and pleasantness ratings (for stimuli used in the main fMRI experiment) (*Table 1*), thereby reducing the possibility that baclofen-induced changes in odor perception could have influenced the imaging results.

## Odor perceptual categorization

Before and after drug administration, subjects participated in an fMRI odor categorization task. On each trial, subjects smelled one of six odors belonging to three categories: citrus (C1 and C2), mint (M1 and M2), and wood (W1 and W2) (*Figure 1b*). Prior to each scanning session, subjects first provided category descriptor ratings (i.e., "how citrusy/minty/woody is odor X?"), as well as pair-wise similarity ratings, for each of the six odors. During the pre-drug session, within-category descriptor ratings were significantly higher than across-category descriptor ratings (simple main effect of category at pre: $F_{1,31}$ = 572.66, P < 0.001; mixed-model ANOVA) (*Figure 3a,b*), in the absence of an interaction between placebo and baclofen groups ($F_{1,30}$ = 0.0019, P = 0.97; *Figure 3b*). Moreover, during the pre-drug session, the within-category odor pairs were rated as significantly more similar than across-category odor pairs (simple main effect of category at pre: $F_{1,31}$ = 125.80, P < 0.001; *Figure 3c*), again without an interaction between groups ($F_{1,30}$ = 0.097, P = 0.76; *Figure 3c*). Finally, based on group averages of similarity ratings, we performed a cluster analysis and found that both placebo and baclofen subjects successfully group the six odors into three classes appropriately in both pre- and post-drug sessions (*Figure 3e*). These data confirm that subjects were highly familiar with the odor categories prior to initiating the experiment.

To quantify categorization performance, we calculated the difference between within-category and across-category descriptor ratings, as well as the difference between within-category and across-category similarity ratings (*Table 1*). There was a main effect of session on categorization

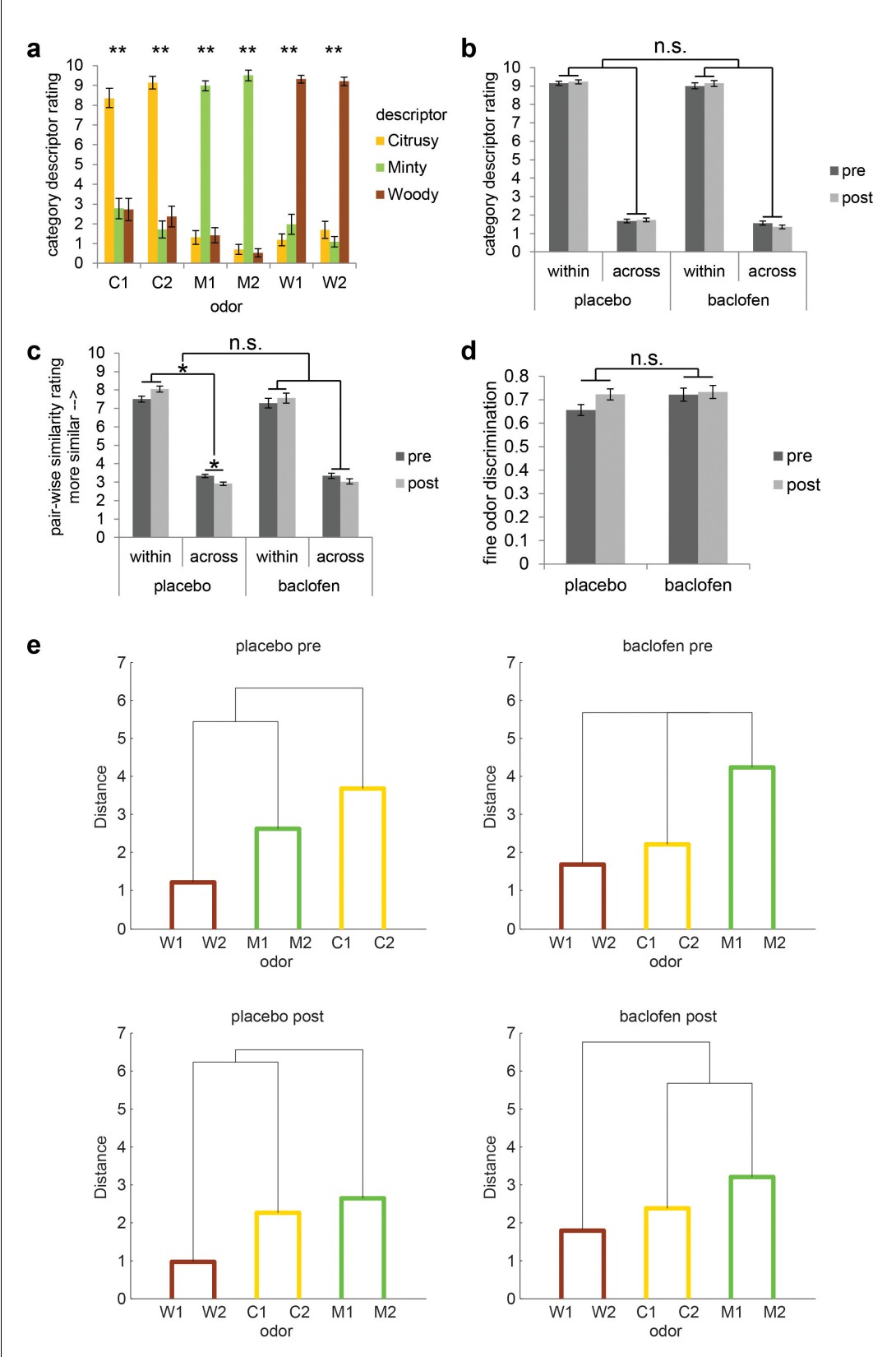

**Figure 3.** Subjects successfully classified odors into their relevant categories. (a) Category descriptor ratings of the six odors (two citrus: C1, C2; two minty: M1, M2; two woody: W1, W2) from all subjects during the pre-drug session (mean ± within-subject s.e.m., n = 32). Repeated-measures ANOVA

*Figure 3 continued on next page*

*Figure 3 continued*

was conducted separately on each odor (** = P < 0.001). Subjects robustly classified the odors into the appropriate perceptual categories (C1: $F_{1.86, 57.77}$ = 31.62; C2: $F_{1.72, 53.40}$ = 74.58; M1: $F_{1.92, 59.61}$ = 144.04; M2: $F_{1.60, 49.46}$ = 373.79; W1: $F_{1.82, 56.33}$ = 140.96; W2: $F_{1.49, 46.10}$ = 166.84; all P's < 0.001). (**b**) Average of category descriptor ratings across odors, sorted by within-category condition and across-category condition in pre- and post-drug sessions for placebo (n = 18) and baclofen (n = 14, mean ± within-subject s.e.m.) groups. (**c**) Pair-wise similarity ratings of within- and across-category odor pairs in pre- and post-drug sessions for placebo and baclofen groups (mean ± within-subject s.e.m.). (**d**) Fine odor discrimination between α- and β-pinene in pre- and post-drug sessions for placebo and baclofen groups (mean ± within-subject s.e.m.). (**e**) Dendrogram plots obtained from a cluster analysis of the average pair-wise similarity ratings for placebo and baclofen subjects during pre- and post-drug sessions showed that both groups sorted the six odors into three categories in both sessions. Shorter distance indicates greater similarity.

showing a general improvement based on similarity ratings ($F_{1,30}$ = 5.65, P = 0.024), probably reflecting a practice effect. However, the session-related changes did not differ between groups ($F_{1,30}$ = 0.30, P = 0.59; *Figure 3c*, *Table 1*). For categorization based on descriptor ratings, there was no main effect of session ($F_{1,30}$ = 0.33, P = 0.57) or interaction between groups ($F_{1,30}$ = 0.30, P = 0.59; *Figure 3b*, *Table 1*). Collectively, these results indicated that baclofen did not affect behavioral measures of odor categorization at the group level.

During fMRI scanning, subjects received occasional 'catch trials' (every 4–8 trials), in which they were prompted to indicate the category of the previously delivered odor. In the pre-drug session, subjects categorized odors with high accuracy (84.4% ± 2.7%, chance level at 33%, $t_{31}$ = 19.37, P < 0.0001). Of note, neither the catch trial accuracies nor reaction times (RT) differed significantly as a function of treatment group from pre- to post-drug session (*Table 1*).

## Category-specific ensemble codes in PPC, OFC, amygdala and pHIP

During the fMRI odor categorization task, the six odors were delivered in a pseudorandom order, and subjects were cued to sniff upon odor delivery. They were asked to pay attention to the quality of the odors throughout the task, and make category judgments during catch trials.

As olfactory information takes the form of distributed patterns of fMRI activity in the human brain (*Howard et al., 2009*; *Wu et al., 2012*), multivariate pattern analyses are well-suited for examining the impact of baclofen on odor pattern recognition. We first used a support vector machine (SVM) classifier to identify brain areas where odor category information is represented, among several anatomically defined regions of interest (ROIs) including piriform cortex, higher-order areas that directly project to piriform (olfactory subregion of OFC, amygdala, entorhinal cortex), and hippocampus (*Figure 4a*). This analysis was conducted for all subjects in the pre-drug session, in order to constrain our investigation of baclofen-induced drug effects to those ROIs that had robust odor category coding at baseline (thus independent of drug administration). We trained the SVM classifier on patterns evoked by one pair of odors belonging to different categories (e.g., C1 vs. M1), and then tested the classifier on patterns evoked by the complementary pair of odors from the same categories (e.g., C2 vs. M2; *Figure 4b*). Importantly, because training and test sets were based on data evoked by different odor identities, significant above-chance decoding is only possible if fMRI patterns encode category information independent of the specific odor identities. Across all subjects in the pre-drug session, we found significant above-chance decoding accuracy in PPC ($t_{31}$ = 2.05, P = 0.024), OFC ($t_{31}$ = 1.96, P = 0.029), amygdala ($t_{31}$ = 3.17, P = 0.0017), and posterior hippocampus (pHIP, $t_{31}$ = 1.90, P = 0.034; *Figure 4c*). All subsequent analyses were constrained to these four regions where fMRI ensemble patterns encode odor category information.

## Baclofen interferes with within-category pattern separation in PPC

In order to characterize the continuous degree of pattern similarity between stimuli (*Nili et al., 2014*), we next used a linear correlation analysis (*Haxby et al., 2001*; *Howard et al., 2009*; *Kriegeskorte et al., 2008*), which provides a more direct assessment of pattern overlap. Specifically, to examine how baclofen alters the categorical organization of odors, we assembled vectors of ensemble pattern activity from all voxels within PPC, and measured the dissimilarity (correlation distances) of pattern vectors evoked by across-category odors (e.g., C1/M1) and within-category odors (e.g., C1/C2, *Figure 5a*). In order to control for within-session and between-session variations that could arise from training, familiarity, increasing boredom, drug effect, scanner drift or other potential

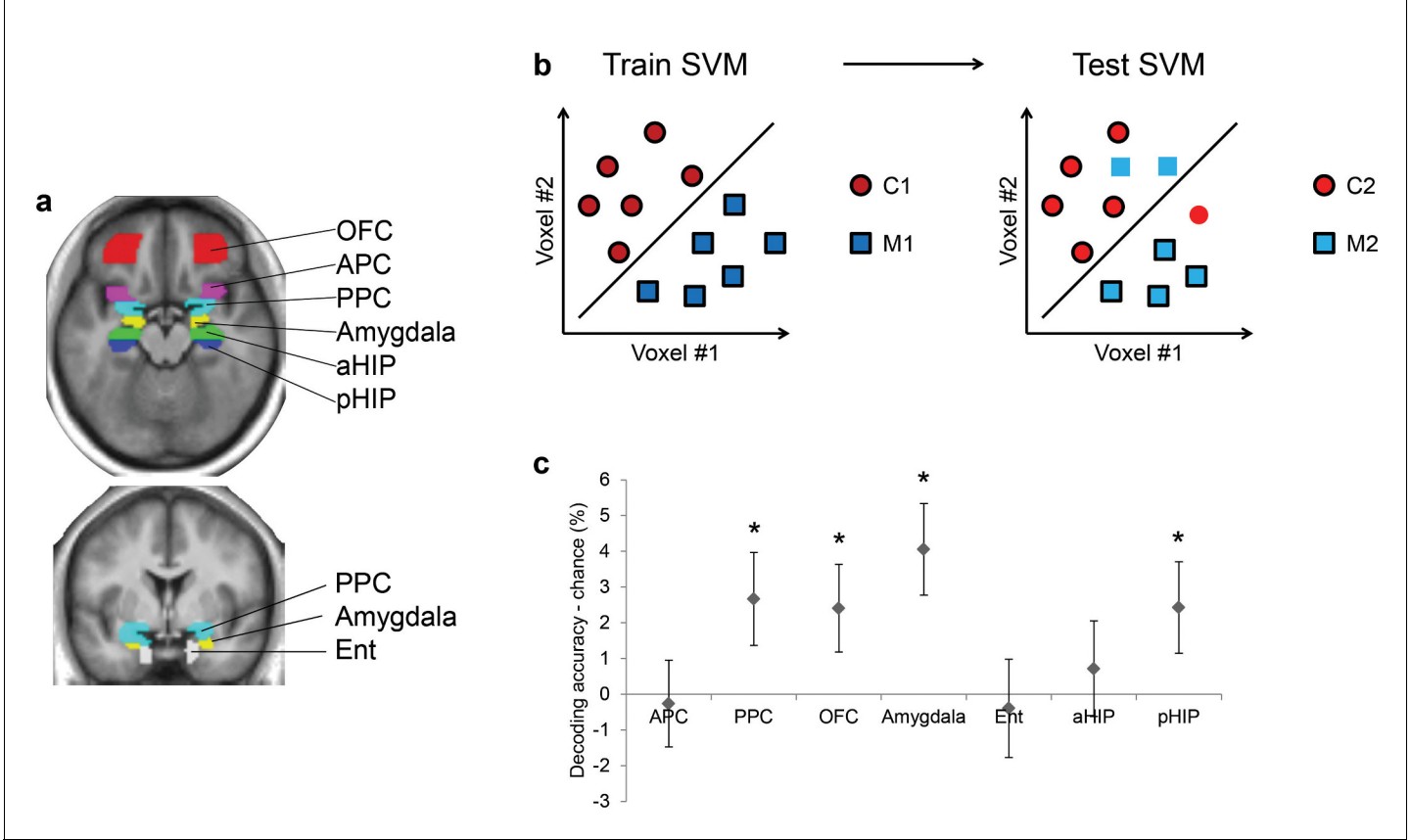

**Figure 4.** Ensemble pattern coding of odor category information at baseline (pre-baclofen session). (a) Axial and coronal slices of the averaged, normalized T1-weighted structural scan from all subjects showing anatomically defined regions of interest. Odor-evoked ensemble patterns across all voxels within a given ROI were used in a two-step multivariate classification analysis. First, we trained a linear SVM on a training data-set (b, left panel) to separate two odors belonging to different categories. Second, odor category coding was assessed in an independent test data-set (b, right panel), specifically by testing how well the SVM classified the other pair of odors from the corresponding categories; here, cross-decoding is only successful if similar patterns code different odors of the same category. (c) Category decoding from all subjects during the pre-drug session showed that classification accuracy in PPC, OFC, amygdala, and pHIP significantly exceeded chance (mean ± between-subject s.e.m., n = 32, *P < 0.05, one-tailed).

artifact, we computed pattern distances between *the same odor* as baseline patterns, and then subtracted these from the within-category and across-category pattern distances. A two-way mixed-model ANOVA on same-odor pattern distances showed that there was no main effects of session or group, or group × session interaction ($F_{1,30} = 1.88$, P = 0.18), indicating that the baseline patterns were consistent across time and group, and were not affected by the drug. We then tested a three-way analysis of variance (ANOVA), with two within-subject factors of session (pre/post) and category type (within-/across-category), and one between-subject factor of drug (placebo/baclofen). This yielded a significant session × category type × drug interaction effect ($F_{1,30} = 5.49$, P = 0.026) in the absence of other main effects or two-way interactions (all P's >0.15), and suggests that baclofen significantly affected the categorical structure of odor pattern representations in PPC.

Based on inspection of the odor-evoked pattern changes in PPC (*Figure 5b*), it is evident that these changes were actually more prominent in the placebo group, and for the within-category condition. To assess these hypotheses, we examined drug-related categorization effects separately in each group. In placebo subjects the interaction of session × category type was significant in PPC ($F_{1,17} = 9.35$, P = 0.0071; repeated-measures ANOVA), whereas no such interaction was identified in the baclofen group ($F_{1,13} = 0.62$, P = 0.45). This effect was driven by a significant increase of the within-category odor distance in the placebo group ($F_{1,17} = 5.23$, P = 0.035), but not in the baclofen group ($F_{1,13} = 2.61$, P = 0.13), with a significant difference between groups ($F_{1,30} = 7.36$, P = 0.011; mixed-model ANOVA, session × group interaction). On the other hand, across-category odor

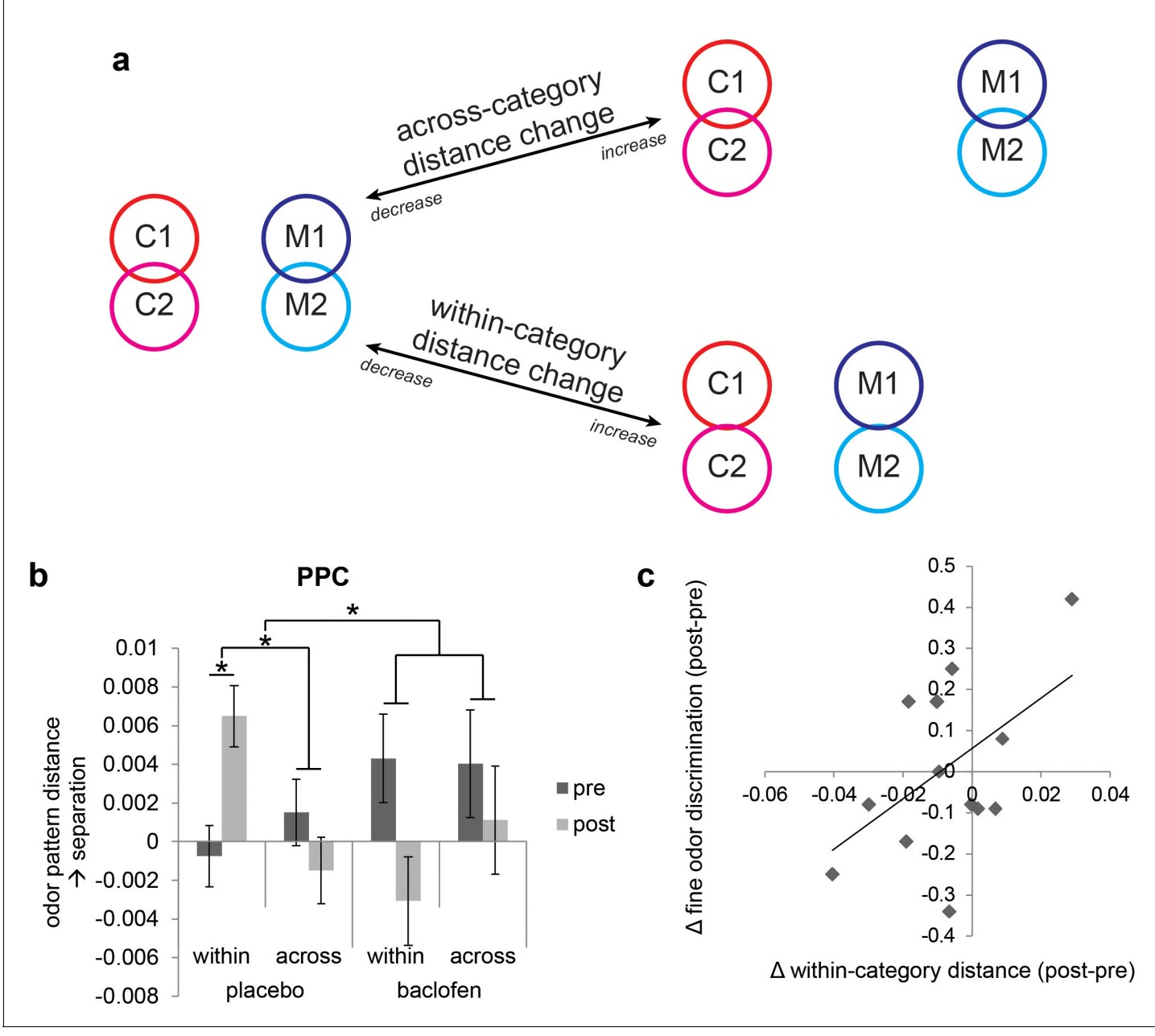

**Figure 5.** Baclofen effect on odor pattern changes in PPC. (a) Schematic illustrating within-category and across-category relationships among categorically organized odors, and how changes of each distance parameter alter the categorical structure. Worse categorization emerges when within-category distances increase or when across-category distances decrease. Better categorization emerges when within-category distances decrease or when across-category distances increase. (b) Odor pattern distance in PPC in pre- and post-drug sessions, sorted by within-category and across-category distances, from placebo (n = 18) and baclofen (n = 14, mean ± within-subject s.e.m.) subjects. Placebo subjects showed increased within-category distances without across-category changes. There was no significant odor distance change in baclofen subjects. (c) A scatterplot showing the correlation between the magnitude of within-category odor pattern separation in PPC and behavioral changes in a fine odor-discrimination task, from pre- to post-drug session (ρ = 0.51, P = 0.031, n = 14, one-tailed). Each diamond represents one baclofen subject. *P<0.05.

distances did not differ for either group (placebo, $F_{1,17}$ = 0.75, P = 0.40; baclofen, $F_{1,13}$ = 0.27, P = 0.61) or between groups ($F_{1,30}$ = 0.00014, P = 0.99, *Figure 5b*).

These results highlight a divergence in PPC pattern representations for odors belonging to the same category, but only in the placebo group. One implication is that repeated exposure to the odors (in absence of drug) induced pattern separation or differentiation, a process that appears to

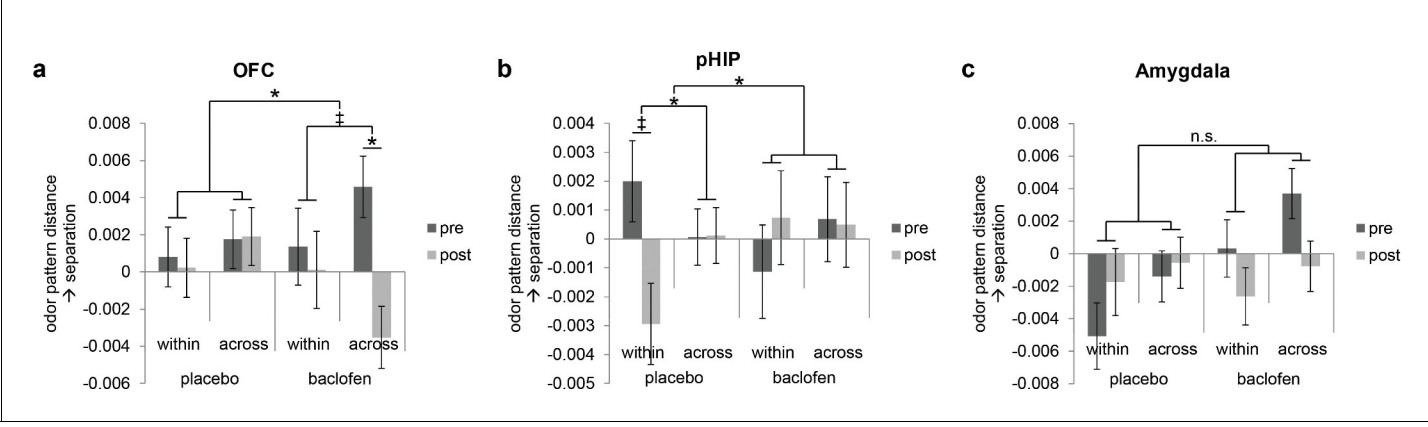

**Figure 6.** Baclofen effect on odor pattern changes in OFC and pHIP. Odor pattern distances in (a) OFC, (b) pHIP, and (c) amygdala in pre- and post-drug sessions, sorted by within-category and across-category distances, for placebo (n = 18) and baclofen (n = 14, mean ± within-subject s.e.m.) subjects. (a) In the baclofen group, across-category distances in OFC decreased significantly without change in within-category distances, leading to disrupted categorical structure. There was no change in the placebo group. (b) In pHIP, the placebo group showed a trend decrease in within-category odor distances without change in across-category distances. There was no significant odor distance change in the baclofen group. (c) In amygdala there was no baclofen effect on the categorical representation of odors. ‡ P<0.1, *P<0.05.

be blocked in the presence of baclofen. Interestingly, this conceptualization – greater pattern separation over time in the control subjects – is in close accordance with an earlier olfactory perceptual learning study from our lab, where prolonged passive exposure to one target odor increased its discriminability from categorically related odors (*Li et al., 2006*). Viewed in this context, it is reasonable to speculate that baclofen interferes with the natural emergence of olfactory pattern separation in PPC, possibly reflecting a disruption in consolidation mechanisms that normally underlie perceptual learning.

If pattern separation in PPC is critical for differentiating categorically related odors, it follows that subjects with greater disruption of PPC pattern separation (as a result of baclofen treatment) should exhibit greater olfactory perceptual deficits. This hypothesis was tested by regressing subject-wise measures of fine odor discrimination (*Figure 3d*) against the magnitude of baclofen-induced pattern changes in PPC. We found a significant correlation between perceptual performance change and the degree of odor-evoked pattern separation in PPC ($\rho = 0.51$, P = 0.031, one-tailed; *Figure 5d*). Thus, subjects with less within-category odor separation in PPC showed greater difficulty in discriminating between odors sharing semantic features.

It is worth considering that because baclofen produced significant sleepiness, the associated tiredness and sedation may have also been associated with a lack of attention on the hardest discriminations. To investigate this possibility, we tested the correlation between pre-post changes in sleep scale ratings and changes in fine odor discrimination performance across baclofen subjects. This relationship was not significant (Spearman $\rho = -0.33$, P = 0.25, n = 14). Likewise, there was no significant correlation between sleep scale changes and within-category pattern distance changes in PPC across baclofen subjects (Spearman $\rho = 0.11$, P = 0.70, n = 14). These data suggest that there was no direct evidence of a systematic link between sleepiness and odor discrimination at the behavioral or neural level.

## Baclofen disrupts category coding in OFC and impedes within-category generalization in pHIP

Because olfactory categorical codes were also identified in OFC, amygdala, and pHIP in the pre-treatment session (*Figure 4*), we also investigated the effects of baclofen on categorical organization of odor ensemble patterns in these regions. Significant three-way interactions of session × category type × drug were found in OFC ($F_{1,30} = 4.48$, P = 0.043) and pHIP ($F_{1,30} = 5.90$, P = 0.021) without other main effects or two-way interactions. No significant interaction was observed in amygdala ($F_{1,30} = 0.047$, P = 0.83; *Figure 6c*).

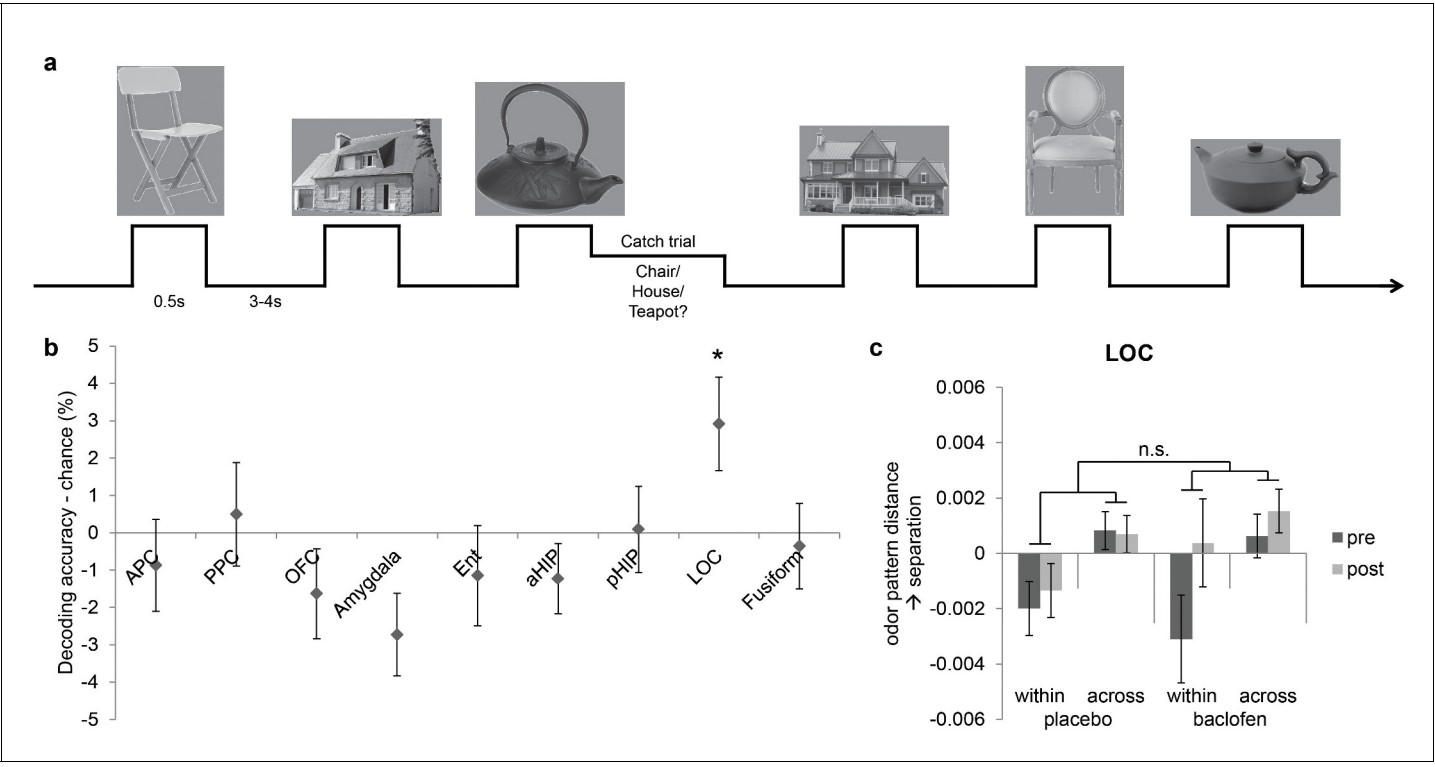

**Figure 7.** Visual control experiment. (**a**) Paradigm of the fMRI visual categorization experiment. Subjects viewed six images belonging to three categories. On catch trials that occasionally followed image presentations, names of the three categories appeared on screen, and subjects indicated the category of the image with a mouse click. (**b**) Visual category decoding from all subjects during the pre-drug session showed that classification accuracy in LOC significantly exceeded chance (*$P = 0.013$, one-tailed). (**c**) The effect of baclofen on visual categorical representations in LOC was not significant ($P = 0.50$).

Following the same approach used for PPC, we next asked how odor category organization in OFC changes in each group. We found that in OFC, the session × category interaction approached significance in the baclofen group ($F_{1,13} = 4.14$, $P = 0.063$), driven by a significant decrease in across-category distances ($F_{1,13} = 5.93$, $P = 0.030$; *Figure 6a*), and leading to a net effect of category disruption (that is, greater convergence of across-category patterns; e.g., C1 and M1 becoming more alike, *Figure 5a*). There was no categorical change in the placebo group (session × category interaction, $F_{1,17} = 0.17$, $P = 0.69$).

By contrast, in pHIP, there was a significant session × category interaction in the placebo group ($F_{1,17} = 7.68$, $P = 0.013$), here driven by a trend decrease in within-category distances ($F_{1,17} = 3.09$, $P = 0.097$, *Figure 6b*), and giving rise to an enhanced categorical structure among odors (that is, greater convergence of within-category patterns; e.g., C1 and C2 becoming more alike, *Figure 5a*). Of note, this profile is opposite to that seen in PPC (*Figure 5b*). On the contrary, there was no categorical change in the baclofen group (session × category interaction, $F_{1,13} = 0.77$, $P = 0.40$).

## Effects of baclofen are specific to olfactory processing

The above findings indicate that baclofen had selective effects on odor category coding in PPC, OFC, and pHIP. However, because baclofen was administered systemically, it remains unclear whether the effects were specific to odor categorization, or merely altered semantic or conceptual processing independently of sensory modality. Therefore, in a parallel fMRI experiment, the same subjects performed a visual categorization task (*Figure 7a*), viewing six images belonging to three categories (chairs, teapots, and houses) and identifying the category on catch trials. There was no effect of baclofen on response accuracies and reaction times (*Table 1*).

We then utilized the same multivariate analysis pipeline to explore the effects of baclofen on visual pattern recognition. First we used the SVM classifier to decode visual category information in

the same ROIs as in the olfactory task. We also included two additional visual ROIs located in lateral occipital complex (LOC) and fusiform gyrus as defined by an independent functional localizer scan, and which are known to be involved in visual object recognition (*Haxby et al., 2001*; *Cox and Savoy, 2003*; *Grill-Spector, 2003*; *Kriegeskorte et al., 2008*). Across all subjects in the pre-drug session, category decoding accuracy was significantly above chance in LOC ($t_{30}$ = 2.33, P = 0.013), but not in fusiform cortex ($t_{30}$ = -1.20, P = 0.88) or in any of the olfactory ROIs (PPC: $t_{30}$ = 0.36, P = 0.72; OFC: $t_{30}$ = -1.36, P = 0.18; amygdala: $t_{30}$ = -2.47, P = 0.99; pHIP: $t_{30}$ = 0.078, P = 0.47; *Figure 7b*). Next, we performed an fMRI pattern correlation analysis to test the drug effect on visual category representations. A three-way session × category type × drug interaction was not significant in LOC ($F_{1,29}$ = 0.46, P = 0.50; *Figure 7c*), suggesting that baclofen did not alter coding of categories in the visual domain.

Finally, we compared the effect of baclofen on categorization between olfactory and visual tasks, and found that the impact of baclofen on category coding in PPC was specific to olfaction. A mixed four-way ANOVA (three within-subject factors of modality, category type, and session; one between-subject factor of group) revealed a significant interaction of modality × category × session × drug ($F_{1,29}$ = 4.41, P = 0.044). Thus, while baclofen blocked within-category separation in PPC, it did not alter the visual categorization compared to placebo (category × session × drug interaction: $F_{1,29}$ = 0.056, P = 0.81). These findings imply that the observed effect of baclofen in PPC was not due to generic changes in semantic processing, nor to non-specific changes in hemodynamic parameters, but instead was due to alterations in information coding in the presence of olfactory inputs.

## Discussion

In this study we investigated the role of piriform associative connections in the neural coding of odor categories. We used the GABA(B) receptor agonist, baclofen, to reduce associative input in the olfactory network while sparing afferent input from the periphery. This pharmacological manipulation, combined with multivariate pattern analysis, enabled us to examine how baclofen treatment alters fMRI pattern representations of odors within and across categories relative to placebo. We found that in PPC, baclofen (compared to placebo) interfered with the emergent pattern separation of odors belonging to the same categorical class. The magnitude of this effect correlated with difficulties in fine-odor discrimination at the perceptual level. In contrast, baclofen disrupted across-category separation in olfactory downstream region of OFC, and impeded within-category generalization in pHIP.

Interestingly, the baclofen effect observed in PPC was opposite to our original prediction that baclofen would simply weaken the boundaries between categories, leading to reduced pattern separation between citrus, mint, and wood odors. Instead, there was increased pattern separation for within-category odors over time, but specifically in the placebo group, likely reflecting perceptual training or stimulus-specific consolidation from the pre- to post-session. The significant group × session interaction implies that the natural process of within-category pattern separation in controls was disrupted in the presence of baclofen. For example, PPC pattern representations of the two citrus odors became more distinct under the control condition, but failed to diverge under baclofen. We speculate that piriform associative input normally supports the separation of patterns corresponding to unique identities of individual odors, especially those sharing perceptual features and associated with the same semantic labels. This mechanism would be compatible with prior work showing that perceptual learning enhances discriminability of within-category odor pairs, with concomitant fMRI changes in PPC as well as OFC (*Li et al., 2006*).

It is worth considering why baclofen had no effect on across-category odor separation in PPC. One plausible explanation is that piriform cortex has the capacity to enhance either pattern separation or completion, as a function of task demands (*Chapuis and Wilson, 2012*; *Li et al., 2008*; *Shakhawat et al., 2014*). Various rodent paradigms of olfactory associative learning have shown that the direction of piriform pattern changes can flexibly match the behavioral requirements for either odor discrimination (i.e., pattern separation) or odor generalization (i.e., pattern completion). In the current experiment, subjects were asked to perform an odor categorization task, in which differences across categories, but not within category, were emphasized. As such, our experimental design might have helped stabilize category-specific differences in PPC, even in the presence of baclofen, though at the expense of within-category odor separation. The fact that categorical

representations of citrus, mint, and wood odors were already highly familiar to the subjects also could have created further stability against across-category pattern changes. Thus, it is fair to say that while we have no evidence to support baclofen-induced disruption of across-category codes, the nature of our task leaves open the possibility that in a different task where categorical distinctions were less relevant, baclofen might have a modulatory influence on across-category differences.

Another potential factor related to the experimental design that might complicate interpretation of the PPC results is a short-term order effect between trials. That is to say, pattern representations of the same odor could differ depending on whether the preceding odor belonged to the same category (category repetition, e.g., C1 preceded by C2) or a different category (non-repetition, e.g., C1 preceded by M1). This repetition factor could induce short-term 'learning' or adaptation effects that involve local synaptic interactions mediated by GABA(B) receptors (*Brenowitz et al., 1998*; *Ziakopoulos et al., 2000*), and thus could be susceptible to baclofen manipulation. Examination of our data indicates that across subjects and pre/post sessions, there were 39.3 ± 0.6 (mean ± SE) category repetition trials as opposed to 121.5 ± 0.9 category non-repetition trials. That the majority (~75%) of trials in our experiment belonged to non-repetition trials suggests that category repetitions would not have had a pronounced effect on the findings. In additional analyses (see Materials and methods), the proportion of category repetition versus non-repetition trials (per imaging run) was not associated with the strength of pattern categorization effects in PPC, and PPC patterns evoked by *the same odor* under repetition and non-repetition conditions were not significantly different. Thus, it is reasonable to conclude that the sequence order of the odors did not have an impact on categorical pattern coding in PPC, either at baseline or in the context of drug.

In contrast to PPC, fMRI patterns in olfactory downstream areas, including OFC and pHIP, showed deficient category coding in the baclofen group. Thus in OFC, the discrete categorical patterns for citrus, mint, and wood became less separated, in the presence of baclofen. In spite of these changes, there was no parallel impact on behavior. Indeed, baclofen had no perceptual effect on categorical discrimination, and we would argue that such a finding would have been unlikely, presumably due to high familiarity and discriminability of odor categories. However, to the extent that the existence of an odor category necessitates an association between an olfactory stimulus and semantic conceptual knowledge, these results are consistent with the recognized integrative role of OFC in guiding olfactory-based behavior. Both animal and human studies have demonstrated that OFC patterns can differentiate between odor objects and categories (*Howard et al., 2009*; *Wu et al., 2012*; *Schoenbaum and Eichenbaum, 1995*; *Critchley and Rolls, 1996*). Moreover, the OFC has been proposed to integrate taste and visual information associated with odor stimuli (*Critchley and Rolls, 1996*; *Gottfried and Dolan, 2003*), encode the reward value of odors (*Howard and Gottfried, 2014*), disambiguate mixtures of categorically dissimilar odors (*Bowman et al., 2012*), and represent olfactory lexical-semantic content (*Olofsson et al., 2014*). Viewed in this context, our results highlight the role of OFC in preserving the perceptual distinctions between different odor categories, likely through its associative access to multimodal and semantic information streams.

The demonstration of olfactory category coding in pHIP, and its vulnerability to baclofen, echoes hippocampal findings in the visual modality (*Seger and Miller, 2010*; *Seger and Peterson, 2013*; *Kumaran and McClelland, 2012*). For example, single-unit recordings from the hippocampus have identified neurons in both humans and monkeys that are able to categorize visual information (*Kreiman et al., 2000*; *Hampson et al., 2004*), and fMRI activity in human hippocampus is selectively increased when memory performance relies on perceptual generalization across stimuli (*Preston et al., 2004*; *Shohamy and Wagner, 2008*). Considered in this framework, the trend effect of *decreased* within-category separation in pHIP in placebo subjects may reflect the role of hippocampus to generalize, or to make inferences, across shared odor features, essentially bringing odors of the same category closer together, and creating more separation between different categories (*Figure 5a*). It is interesting to note that both piriform cortex and hippocampus have long been regarded as canonical models of autoassociative networks where pattern separation and pattern completion computations can be flexibly achieved (*Yassa and Stark, 2011*; *Bekkers and Suzuki, 2013*; *Wilson, 2009*; *Leutgeb and Leutgeb, 2007*; *Hunsaker and Kesner, 2013*; *LaRocque et al., 2013*; *Eichenbaum et al., 2007*). That the effect of baclofen was to impede within-category *discrimination* in PPC, while simultaneously impeding within-category *generalization* in pHIP, highlights a unique functional difference between these two anatomically homologous regions, and may help

bring new mechanistic understanding of the contributions of piriform cortex, hippocampus, and piriform-hippocampal interactions to human olfactory processing and perception.

Our behavioral data indicate that the 50-mg baclofen dose did not impair general cognition or olfactory perceptual performance, suggesting that off-target effects of the drug were minimal, other than a modest effect on subjective sleepiness that did not interfere with online task accuracy or response times. While it is possible that the 50-mg dose may not have been potent enough to exert a physiological effect, the study medication schedule was similar to those used in other human studies that administered baclofen to induce reliable changes in brain activity or behavior (*Terrier et al., 2011*; *Franklin et al., 2012*; *Young et al., 2014*; *Franklin et al., 2011*). In our study, we did not find evidence of drug effect on odor categorization behavior, in spite of significant changes in fMRI pattern representations. There are at least three possibilities to account for this discrepancy. One possibility is that our behavioral tests were simply not sensitive enough to detect changes in perceptual performance. Across both placebo and baclofen groups, there was a general trend towards improved performance, likely reflecting effects of training and exposure. As such, any further subtle effect of baclofen on perception may not have emerged beyond these training effects per se. Related to this, even if the three-way forced-choice pinene triangle test was arguably the most 'sensitive' or difficult test of odor discrimination, this test might not have revealed a significant change if the perceptual learning effects had been confined to those odors presented repeatedly during the fMRI. A second possibility is that because all of the subjects were explicitly informed of the categorical features of the odor stimuli, there may have been an implicit tendency to anchor their perceptual responses to semantic categorical attributes. Moreover, even at baseline, all of the odors were easy to discriminate and highly familiar, and subjects were regularly called upon to make category judgments of the odors throughout the experiment. Thus, even though there was reorganization of categorical representations in PPC, these factors could have obscured our ability to observe perceptual plasticity across the set of odors. Finally, while changes in piriform odor representations might have induced parallel changes in perception, it is possible that other brain areas would be able to compensate for these perturbations, helping to stabilize olfactory perceptual performance. For example, in the placebo group, increased within-category pattern separation in PPC (*Figure 5b*) would be counteracted by decreased within-category pattern separation in pHIP (*Figure 6b*), resulting in no detectable change at the behavioral level.

One potential issue is that baclofen can also target GABA(B) receptors that have been identified in area CA1 of the hippocampus, influencing visual object recognition and memory (*Lanthorn and Cotman, 1981*; *Ault and Nadler, 1982*). Therefore, to establish that our findings were specific to the olfactory system, and to ensure that baclofen did not disrupt general semantic processing and object categorization, subjects also performed a visual categorization fMRI task in which they viewed pictures rather than smelled odors. This control study confirmed that our pharmacological manipulation induced both regional and modality specificity, thus ruling out possible confounds such as altered global attention, arousal, or hemodynamic reactivity. As an added way to minimize mere drug effects, we explicitly focused our imaging analyses on the interactions between group (baclofen/placebo), session (pre/post), and category level (within/across), effectively cancelling out any other session-related confounds.

An unavoidable limitation of this study was that baclofen was administered systemically. While our findings demonstrate regionally selective treatment effects in PPC, it is not possible to confirm that these changes were due to the direct action of baclofen solely at piriform cortex. There are at least three mechanisms by which baclofen could affect categorization in the olfactory network, none of which are mutually exclusive. First, baclofen might directly target the layer 1b synapses in piriform cortex where associative intracortical and extracortical inputs predominate. This would most closely mirror what has been tested using focal baclofen injections in animal models (*Poo and Isaacson, 2011*; *Barnes and Wilson, 2014*), and would underscore the idea that categorical odor representations rely on associative information processing within this layer of piriform cortex. Second, baclofen might target neurons in OFC, entorhinal cortex, and other associative brain areas that project onto piriform cortex. Given that the fMRI BOLD response is thought to reflect local dendritic processing and population activity (*Logothetis and Wandell, 2004*; *Hipp and Siegel, 2015*), our findings could reflect a distant action of baclofen on OFC (or other areas), which in turn alters distributed fMRI patterns measured in piriform cortex. Third, the changes seen in PPC could theoretically have arisen in the olfactory bulb, where GABA(B) receptors have also been described (*Nickell et al., 1994*;

*Palouzier-Paulignan et al., 2002*; *Okutani et al., 2003*; *Wachowiak et al., 2005*; *Aroniadou-Anderjaska et al., 2000*; *Isaacson and Vitten, 2003*; *Karpuk and Hayar, 2008*). In this instance, one might have predicted a more profound olfactory perceptual deficit, including impairments of odor threshold, identification, and perceived intensity, though such a profile was not found in our study. Irrespective of the specific mechanism or mechanisms, these findings establish a critical role of the GABA(B) receptor in modulating categorical representations in PPC and OFC, with specificity for the olfactory modality.

In summary, our study provides a foundation for understanding the contribution of afferent and associative inputs to odor categorical perception in the human brain. Of note, this work forms a counterpoint to an earlier study from our lab in which subjects underwent a 7-day period of odor deprivation (*Wu et al., 2012*): by reducing olfactory afferent input, we were able to show that multi-variate pattern representations of odor category were selectively altered in OFC, without any pattern-based changes observed in PPC. By comparison, in the current study, we were able to test the inverse manipulation, using baclofen to reduce olfactory associative input. In this instance, we again observed a disruption of odor categorization in OFC, but also an interference of session-related pattern changes in PPC and pHIP. The fact that within-category pattern changes in PPC were complementary to those in pHIP, in conjunction with the different course of across-category changes in OFC, underscores the idea that odor categorization is a dynamic process involving multiple stages of an extended olfactory network. We surmise that under normal conditions, the ability to refine discriminability of within-category odors in PPC through experience helps to improve perceptual acuity and decision making, and to prevent perceptual generalization from becoming maladaptive. With the interruption of associative input, in the setting of experimental baclofen or even perhaps as the consequence of a neurological disorder, within-category boundaries can become obscured, leading to perceptual over-generalization that can result in detrimental choices. As such, our findings may point toward an important mechanism by which associative networks regulate perceptual processing. Whether such mechanisms are restricted to the olfactory modality, or apply more widely across different sensory systems, remains to be determined.

## Materials and methods

### Subjects

We obtained informed consent from 36 subjects (mean age, 25 years; 18 baclofen and 18 placebo, with equal numbers of men and women in each group) to participate in this study, which was approved by the Northwestern University Institutional Review Board. Subjects were right-handed nonsmokers with no history of significant medical illness, psychiatric disorder, or olfactory dysfunction. Four female baclofen subjects were excluded from the results due to either excessive movement or falling asleep in the scanner, leaving a total of 14 baclofen subjects.

### Study design

Prior to the main experiment, we conducted a screening session to ensure that subjects had normal olfactory abilities and were able to categorize odors reliably. Subjects rated intensity, pleasantness, and familiarity for six odors belonging to three categories. Each odor was presented three times. Familiarity was rated on a visual analog scale (VAS) with end-points of 'extremely unfamiliar' and 'extremely familiar', and the cursor was reset at the mid-point on every trial. For analysis, the VAS ratings were scaled to a range from 0 (extremely unfamiliar) to 10 (extremely familiar). The mean familiarity rating of the six odors across all subjects was $7.68 \pm 0.20$, suggesting that subjects found the odor set to be relatively familiar. There was also no difference in odor familiarity ratings between subjects assigned to the placebo and baclofen groups (two-sample t-test, $t_{30}$ = -1.18, P = 0.25). Additionally, during the screening session, and following the odor ratings, subjects were asked to smell the six odors from glass bottles and to sort them into three categories. All subjects enrolled in the study were able to sort the odors appropriately into the three categories. In this manner, subjects were pre-exposed to the odors, found them to be familiar, and could associate them with categorical knowledge.

The total length of the experiment spanned 5 consecutive days. Following enrollment, subjects were randomly assigned to the baclofen (n = 14) or placebo (n = 18) group by the research

pharmacy at Northwestern Memorial Hospital. Experimenters and subjects were both blinded to these assignments. Subjects took 10 mg of baclofen or placebo on the first day and progressively increased the dosage by 10 mg per day to reach 50 mg at day 5. On day 1 before drug administration, subjects underwent pre-drug baseline tests including cognition, olfactory psychophysics, and fMRI imaging measures. On day 5 after medication, subjects completed post-drug tests which were the same as the pre-drug session.

## Odor stimuli and delivery

Six odorants were used in the fMRI odor categorization experiment and included two 'citrus' smells (R-(+)-limonene and Citral), two 'mint' smells (L-Menthol and Methyl Salicylate), and two 'wood' smells (Cedrol and Vetiver Acetate). For the fine odor discrimination task outside the scanner, two perceptually similar isomers, α- and β-pinene (5% diluted in mineral oil), were used in an olfactory three-way forced choice triangular task. Odors were delivered using a custom-built olfactometer. In this system, clean air or odorized air was directed towards subjects (wearing a nasal mask) via Teflon tubing at a rate of 3 L/min.

## General cognitive measures

On days 1 and 5, subjects were tested on four cognitive measures before olfactory testing and fMRI scanning: (1) Mini-mental state examination (MMSE), a short questionnaire used to measure cognition impairment (*Folstein et al., 1975*); (2) an auditory digit span test (in forward and backward order) to assess short-term memory; (3) Trail Making Test B as a measure of visual attention and cognitive flexibility (*Bowie and Harvey, 2006*); and (4) subjective report of degree of alertness using the Stanford Sleepiness Scale (SSS) (*Hoddes et al., 1973*), which ranges from "Feeling active, vital, alert, or wide awake" (1 point) to "No longer fighting sleep, sleep onset soon; having dream-like thoughts" (7 points).

## Olfactory psychophysical measures

Four behavioral measures were tested outside of the scanner. (1) Odor detection thresholds and (2) odor identification ability were assessed using Sniffin' Sticks (Burghart) and the University of Pennsylvania Smell Identification Test (UPSIT, Sensonics), respectively (*Doty et al., 1984*; *Hummel et al., 1997*). (3) A triangular odor discrimination task was performed to assess the ability to discriminate α- and β-pinene (*Li et al., 2008*). (4) For the six odorants used in the fMRI odor categorization experiment, visual analog ratings of odor intensity (anchors, 'undetectable' and 'extremely intense'), pleasantness (anchors, 'dislike", 'neutral', and 'like'), pair-wise similarity of odor quality (anchors, 'not alike at all' and 'identical') (*Howard et al., 2009*) were collected. Subjects also rated the applicability of descriptors of the three categories (citrus, mint and wood) with anchors ('not at all' and 'extremely citrusy/minty/woody').

## fMRI olfactory and visual categorization tasks

Subjects underwent an odor categorization task designed to assess the multivoxel pattern specificity of odor-evoked fMRI activity across pre- and post-drug sessions. The task was divided into six 8-min runs of 28 trials each, during which the six odors were presented for 4 or 5 trials (depending on the run). On each trial, subjects were presented a visual sniff cue prompting them to sniff. Odor stimuli were presented for 1.5 s, with a 13-s stimulus-onset asynchrony (SOA). Each odor was presented 28 times in pseudorandom order. Four out of the 28 trials in each run were randomly chosen as 'catch trials', where subjects were asked to indicate the category of the received odor with a mouse click. The catch trials were not included in the fMRI pattern analysis. The total task lasted for 48 min.

Subjects also performed a visual categorization task which was parallel to the olfactory version with the equivalent number of trials and runs, and visual and olfactory runs were interleaved. On each trial, an image (from a total of six possible images, *Figure 7a*) was presented for 0.5 s, with a jittered interval of 3–4 s between trials. The visual fMRI data were absent from 1 male placebo subject due to technical problems during the experiment.

### fMRI visual ROI localizer scan

A separate functional localizer scan was performed to identify regions of image-evoked activity to be used in the visual pattern analysis. This scan was done in the pre-drug session, in which subjects were shown seven 20-s blocks of images (0.3 s presentation and 0.7 s inter-stimulus interval) with 20-s resting gaps between blocks. Each block contained one of six object categories (chairs, houses, teapots, cars, keys, and scissors) or scrambled version of the same images. The scrambled images were created by dividing the images into $20 \times 20$ unit grids and shuffling the units. During the image presentation blocks, subjects performed a one-back detection task by pressing a button to maintain their focus and attention.

### Respiratory monitoring and analysis

Breathing behavior was monitored during olfactory scanning with a spirometer (affixed to the nasal mask) measuring the flow of air during inhalation and exhalation. Respiration signals from each run were first smoothed and then scaled to have a mean of 0 and standard deviation of 1. The cued sniff waveforms were extracted from each trial, and inhalation peak flow, duration, and volume were computed. In the pre-drug session, there were no systematic differences in peak flow ($F_{3.4,105.52} = 1.44$, $P = 0.23$, repeated measures ANOVA) or duration ($F_{3.97,123.18} = 0.89$, $P = 0.47$) across odors, but the inhalation volumes were different ($F_{3.93,121.88} = 3.27$, $P = 0.014$). Therefore the inhalation volume was included in the fMRI analysis as a nuisance regressor (see below).

### fMRI data acquisition

Gradient-echo T2*-weighted echoplanar images were acquired with a Siemens Trio 3T scanner using parallel imaging and a 12-channel head-coil (repetition time, 2.3 s; echo time, 20 ms; matrix size, $128 \times 120$ voxels; field-of-view, $220 \times 206$ mm; in-plane resolution, $1.72 \times 1.72$ mm; slice thickness, 2 mm; gap, 1 mm). A 1 mm$^3$ T1-weighted MRI scan was also obtained for defining anatomical regions of interest (ROIs).

### fMRI pre-processing

fMRI data were pre-processed with SPM8 software (http://www.fil.ion.ucl.ac.uk/spm/). All functional images across pre- and post-drug sessions were spatially realigned to the first scan of the first run to correct for head movement. The T1 structural image was also co-registered to the mean aligned functional image. Realigned functional images were then normalized into a standard space using the transformation parameter from each individual's T1-weighted scan to the standard T1 template. For multivariate fMRI analysis of olfactory and visual categorization scans, we did not perform subsequent spatial smoothing in order to preserve the voxel-wise fidelity of the signal. Images from visual localizer scans were smoothed for generating functional visual object recognition ROIs.

### fMRI data analysis

General linear model

For each subject, a general linear model (GLM) was specified for each categorization scanning run in pre- or post-drug sessions from the spatially aligned, normalized, and unsmoothed fMRI data. An event-related GLM was created by modeling sniff or image onset times of each condition independently with stick (delta) functions, and then convolving with a canonical hemodynamic response function (HRF to generate 6 regressors of interest. This model also included one regressor of no interest (catch trial onsets), six movement parameters derived from spatial realignment, and one sniff parameter (for olfactory scans) derived from inhalation volume convolved with HRF and orthogonalized with the main odor events. The data were high-pass filtered (cutoff period of 128 s) to remove signal drifts, and temporal autocorrelation was adjusted using an AR(1) process. Voxel-wise, odor/image-specific β values were then estimated.

To localize visual object recognition ROIs, a block-design GLM was built on normalized and smoothed localizer scans by modeling each image block onset with a boxcar predictor convolved with HRF. Voxel-wise, condition-specific β values were estimated for object and scramble conditions. Subsequently, the contrast of object > scramble from each subject was entered into a one-sample t-test model at the group level to look for voxels that responded more strongly to objects than scrambles. Continuous clusters of voxels in bilateral LOC (p<0.00001, peak coordinate: right LOC, x

= 44, y = -76, z = -6; left LOC, -44, -80, -2; MNI coordinate space) and fusiform cortex (p<0.001, right: 38, -34, -22; left: -40, -52, -20) were selected as visual ROIs.

## Multivariate pattern analysis

Following GLM estimation, we extracted 36 β pattern vectors (one vector for each of the 6 odors/ images and each of the 6 runs) from all voxels within anatomically defined bilateral ROIs, manually drawn on the mean image of normalized T1 scans of all subjects, using MRIcron software (http:// www.mccauslandcenter.sc.edu/mricro/mricron/). A human brain atlas was used to help delineate the anatomical borders of anterior and posterior piriform cortex (APC and PPC), amygdala, and hippocampus (*Mai et al., 1997*). The boundary of anterior and posterior hippocampus was delineated at the uncal apex (y = -21 in MNI space) (*Poppenk et al., 2013*). The delineation of olfactory OFC was guided by an olfactory fMRI meta-analysis (*Gottfried and Zald, 2005*). The entorhinal cortex was drawn with reference to an MR volumetric analysis of the human entorhinal cortex (Insausti et al., undefined). Visual ROIs of LOC and fusiform were defined by the independent functional localizer scan, as described above.

For multivariate pattern analysis, because we focused on information encoded in distributed fMRI patterns, the pattern vectors from the left and right hemisphere of each ROI were individually scaled to have a mean of 0 and standard deviation of 1, and then concatenated together for bilateral ROI pattern analysis. This assures that the mean signal and any lateralization of activity does not account for information coding.

The LIBSVM (Library for Support Vector Machines, https://www.csie.ntu.edu.tw/~cjlin/libsvm/) implementation was used to decode category information from patterns within a given ROI at baseline (*Chang and Lin, 2011*). We trained the SVM classifier to separate pairs of odors of different categories (e.g. C1 vs. M1) using all six runs, and then tested the SVM by classifying odor patterns of corresponding categories but different identities (C2 vs. M2). Because the training set and testing set contain odors of different identities, significant above-chance decoding is only possible when the patterns code category-specific information independent of the identities.

Based on the regions identified by the SVM classifier in the baseline (pre-drug) session, pattern correlation analysis was then applied to these data, in an effort to characterize changes in pattern separation from pre- to post-drug sessions. Pattern dissimilarity (correlation distance) between presented stimuli was estimated by computing the linear correlation coefficient between vectors of β patterns across pairs of different runs and subtracting from 1 (thus, maximal similarity = minimal distance = 0). All possible pair-wise comparisons were calculated, and then averaged across same-odor distances (e.g., C1 in run 1 vs. C1 in run 2), within-category distances (e.g., C1 in run 1 vs. C2 in run 2), and across-category distances (e.g., C1 in run 1 vs. M1 in run 2). To control for potential drift and variations from one odor to the next in a session or between the sessions, we subtracted same-odor correlation distances from within-category and across-category correlation distances. The adjusted within- and across-category odor distances were used in group-level statistical analysis to test for drug effect.

## Analysis of trial-wise order effects on odor patterns in PPC

To examine whether category repetitions (from one trial to the next) might introduce short-term perceptual biases of familiarity, working memory, or adaptation, which could otherwise complicate interpretation of the fMRI pattern findings in PPC, we conducted two additional control analyses.

First, we re-examined the original fMRI pattern dataset and MVPA model design. Here we took advantage of the fact that each of the scanning runs (28 trials per run) contained different proportions of odor category repetitions and non-repetitions. Our prediction was that runs containing greater numbers of category repetition trials (range across runs, 0%–52.2% repetition trials) would systematically bias pattern categorization effects in PPC. In this analysis, performed in each subject, the percent of category repetition trials was calculated for each odor in each run, and then these values were regressed against the corresponding measure of PPC categorization strength (indexed as across-category minus within-category pattern distances) for each run. The resulting correlation values from each subject were then submitted to one-sample t-tests or repeated-measures ANOVA across subjects. In the pre-drug session, the correlation coefficient across all subjects did not significantly differ from zero (r = -0.015 ± 0.024, $t_{31}$ = -0.62, P = 0.54). There was also no difference

between groups in the pre-drug session (one-way ANOVA, $F_{1,30} = 0.097$, $P = 0.76$), or in the interaction between session (pre/post) and group (mixed two-way ANOVA, $F_{1,30} = 0.58$, $P = 0.45$).

Second, we conducted a new MVPA analysis in which single-trial odor patterns were extracted from the 'raw' fMRI responses in PPC. This procedure enabled us to directly separate the odor trials into category repetition and non-repetition conditions, and offered a way to test whether patterns evoked by a given odor under the repetition condition were significantly different from the non-repetition condition. Data pre-processing included extraction of fMRI signal intensity from each voxel in PPC for each scanning run, temporal de-trending with a high pass filter (cut-off frequency at 0.01 Hz), and assembly into linear vectors of trial-specific voxel activity. Trial-by-trial pattern vectors were then sorted into 'repetition' and 'non-repetition' conditions for each of the 6 odors. For example, C1 trials preceded by a same-category trial (either C1 or C2) were assigned as repetition (rep) conditions; C1 trials preceded by a different-category trial (either M1, M2, W1, or W2) were assigned as non-repetition (non-rep) conditions. This procedure was independently applied to each of the 6 odors in turn. Then, for a given odor (e.g., C1), pair-wise correlation coefficients were calculated (1) between all pairwise repetition conditions (e.g., $C1_{rep}$ versus $C1_{rep}$), and (2) between repetition and non-repetition conditions (e.g., $C1_{rep}$ versus $C1_{non-rep}$). All possible pair-wise correlations were calculated, and then averaged across odors and comparisons. The main prediction was that if trial repetition had an impact on pattern category coding in PPC, then the correlation in (1) would significantly differ from the correlation in (2). Group-wise statistical analysis indicated that there was no difference between the two types of correlations in PPC in the pre-drug session (paired-sample t-test, $t_{31} = 0.42$, $P = 0.68$), nor did a mixed three-way ANOVA reveal any difference in the interaction of order condition (within/between) × session × group (two within-subject factors of order condition and session; one between-subject factor of group; $F_{1,30} = 0.51$, $P = 0.48$).

## Statistics

Results are shown as mean ± s.e.m. for subjects and sessions. For determining category encoding regions, we used one-tailed t tests to compare decoding accuracy to chance. To test for drug effects on behavior and fMRI patterns, we used a mixed-model three-way ANOVA, with one between-group 'drug' factor (placebo/baclofen) and two repeated-measures within-subject factors of 'session' (pre/post) and 'category type' (within/across). Here the critical contrast was the group × session × category interaction, with post-hoc analysis of the simple effects where appropriate. Pearson's linear correlation coefficient was calculated for the correlation analysis of behavioral and fMRI pattern data across subjects. Significance threshold was set at p<0.05, two-tailed, unless otherwise stated.

## Acknowledgements

We would like to thank John Detre at the University of Pennsylvania and Yu Fen Chen at Northwestern University for initial help with protocol development, and Thorsten Kahnt for helpful suggestions. This work was supported by National Institute on Aging (NIA) Grant T32AG20506 (Institutional predoctoral training program) to XB, bioscience internship grant from the École Normale Supérieure (France) to LLGR, and National Institute on Deafness and Other Communication Disorders (NIDCD) Grant R01DC010014 to JAG.

## Additional information

### Funding

| Funder | Grant reference number | Author |
| --- | --- | --- |
| National Institute on Deafness and Other Communication Disorders | R01DC010014 | Jay Gottfried |
| National Institute on Aging | Institutional predoctoral training grant, T32AG20506 | Xiaojun Bao |
| École Normale Supérieure | Bioscience Internship | Louise LG Raguet |

The funders had no role in study design, data collection and interpretation, or the decision to submit the work for publication.

## Author contributions

XB, Conception and design, Acquisition of data, Analysis and interpretation of data, Drafting or revising the article; LLGR, SMC, Conception and design, Acquisition of data; JDH, Analysis and interpretation of data, Drafting or revising the article; JAG, Conception and design, Analysis and interpretation of data, Drafting or revising the article

## Author ORCIDs

Xiaojun Bao, http://orcid.org/0000-0002-8310-4141

## Ethics

Human subjects: Informed consent was obtained from subjects to participate in this study, which was approved by the Northwestern University Institutional Review Board.

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
