## [Decision Letter]

Thank you for submitting your work entitled "The Role of Piriform Associative Connections in Odor Categorization" for consideration by *eLife*. Your article has been favorably evaluated by Timothy Behrens (Senior editor) and three reviewers, one of whom, (Lila Davachi) is a member of our Board of Reviewing Editors.

The following individuals involved in review of your submission have agreed to reveal their identity: Geoff Schoenbuam (peer reviewer); Brice Kuhl (peer reviewer).

The reviewers have discussed the reviews with one another and the Reviewing Editor has drafted this decision to help you prepare a revised submission.

SUMMARY:

In this paper the authors test effects of baclofen on BOLD response during odor exposure and categorization. The authors tested olfactory perception and scanned during odor exposure before and after loading on baclofen. Baclofen is hypothesized to selectively affect associative connections within piriform cortex based on in vitro animal studies. They found no group effects of baclofen on any cognitive measures, however they found that baclofen reduced within-category pattern separation on an MVPA analysis of PPC. This change correlated with reduced odor discrimination ability.

Essential revisions:

Thank you for your submission. All three reviewers (including myself) found the work to be informative, novel and compelling. The manuscript was clearly written and the results potentially pointing to a role of GABA in pattern separation of related odors in olfactory cortex. All the reviewers, however, also felt that certain aspects of the data analyses and reporting needed clarification. The critical results felt 'hidden' in the presentation of 'subtracted' classification results. In addition, at least one reviewer requested some additional analyses. I don't anticipate the additional analyses (order effects? pre- versus post-effects) will be too burdensome but, of course, the new results may reveal something that was not appreciated and, hence, the paper will go out for re-review. I am including a list of points that should be addressed in full if you choose to submit a revision.

1) The pattern of data needs further explanation and unpacking. For instance, in Figure 5 – in support of their conclusion that baclofen leads to a reduction in pattern separation for within category odors (and not across category odors), they show changes in pattern distance for both placebo and drug. Is it possible that the effects noted are driven by greater pattern separation in controls over time? (which I think is what is being graphed)? This would lead one to a different interpretation about the results – that somehow after initial exposure without drug, the brain continues to differentiate – more of a consolidation effect? I may be confused however – but to help the reader better understand the pattern of results, it would be important for *all* the data to be plotted, the actual pattern distance Pre and Post separately instead of just the difference scores.

2) Also, since there was not a significant main effect on any measure of odor perception or discrimination – indeed it looks as if everyone is doing better in the pre- versus the post assessment – how is this reconciled with the relationship between piriform odor representations and behavior?

3) I am a little concerned about the fact that differences are only seen for the fine-grain discriminations (within-category). Since the drug did produce significant sleepiness, is it possible that the tiredness and sedation associated with the drug may have also been associated with a lack of attention to the hardest discriminations? Their results still show that piriform cortex is not conducting this discrimination well but that would be an entirely different spin on the results.

Beyond that, I think the results are interesting. Since differences are seen across areas in the pattern of changes with Baclofen, it is easy to rule out any general effects of drug.

4) I think there are several confounds that are important to address. One is simply an order effect. Does the analysis take into account effects of prior odor? For example, one might expect that the MVPA would be very different to C1 after C2 than after W1 for example. This sort of short term "learning" or familiarity effect may be highly dependent on local synaptic interactions and thus susceptible to pharmacological manipulation. I would like to know that this ordering effect is either not present (or controlled for) and more importantly that the drug effect is independent of any ordering effect. Or not. If there is such an interaction, there is nothing wrong with that, but it would be an important factor in the interpretation.

5) A second confound is the novelty/familiarity across scans and the effect of drug. If I understand the design correctly, the subjects saw the odors for the first time pre-drug, then for the second time, post-drug. So the odors are novel and then familiar, if they are the same odors. Even if new odors are used the second time around, then the categories are still somewhat novel and then familiar. Either way, novelty is potentially confounded with drug effect. If the odors were not pre-exposed somehow in a serious way on a prior day (i.e. not same day as scanned), then I think this is important to note. This also brings into play effects of drug on consolidation of the initial exposure experience. I actually first thought of this when looking at the data on hippocampus. There is an effect of drug there, but really it looks like it is due to a loss of the normal decline in within-category pattern separation in the hippocampal BOLD response no? It seems to me that this may reflect a hippocampal generalization, essentially bringing closer together odors that are normally spaced far apart (i.e. diff categories) by virtue of their co-occurrence in the scanner/testing situation. Anyway again some consideration of this may be in order.

6) I think just changes with time both within and across sessions are not being considered properly. Thus you might have changes in pattern from one odor to the next in a session or between the sessions (even independent of novelty). Eliminating this factor might give you a much clearer picture of what is happening it seems to me. And the authors have the perfect data to address this in the form of "same odor" contrasts. That is, if they analyzed the patterns evoked by the same odor from trial to trial and then took that as a baseline, they should control for such drift. Additionally, the classifier would perform a bit better at this than at within category comparisons. This would validate that there actually is a price to be paid for changing the odor within category. And again, it would be worthwhile to see that there was no effect of drug on this measure. Or to identify if there is an effect, since again this would impact the interpretation I think.

7) I think some comparisons should be done by ANOVA or similar non-parametric multifactorial comparison. I noted one place in my review – the behavioral data – but I think possibly some of the other contrasts could be done this way, comparing group effects X within/across category for example.

8) Some of the critical data feel hidden. For example, at the end of the first paragraph of the subsection “Baclofen disrupts within-category odor discrimination in PPC” –: the three-way interaction between session (pre/post), category (within/between) and drug (placebo/baclofen) is obviously a critical test but the authors don't fully lay out the data for inspection. That is, none of the figures show the full set of 8 data points (e.g., bars) in a single figure. Rather, the effect is unpacked via other analyses that involve some subtraction of conditions from one another. In particular, I think the "categorization index" might be eliminated altogether (e.g., Figure 5). Even for 5C (which uses a pre-post subtraction), I think it would be better to start with a figure that shows all of the data (i.e., the 8 bars) and then show the subtracted data. Showing the full set of data would also allow the reader to actually see the relative size of within category and between category distance effects (this is currently 'hidden' because all measures that are shown are pre minus post).

9) In the last paragraph of the Results: “[…] while baclofen enhanced odor category coding in PPC […]” I believe the authors are referring to the fact that the change in "categorization index" was higher (more positive) in the baclofen group than the placebo group. But, as I note above, I find the categorization index to be a bit confusing and potentially misleading. It reflects the fact that, in PPC, the distance of within-category exemplars decreased relative to across-category exemplars. But I don't see how this is equivalent to "enhanced odor category coding." As shown in Figure 5, between-category distances did not change at all as a function of drug condition. This comes up again in the Discussion: "the net effect of baclofen was to enhance overall categorization in PPC […] “but that conclusion does not obviously follow from the findings.

10) One limitation is that we don't know how the task demands influenced the results. As the authors note in the Discussion (third paragraph), the fact that subjects were tasked with making category-level decisions, but not within-category discriminations, during fMRI scanning may well have been a factor in the observed results. This limitation is not necessarily a critical problem, but it does raise the possibility that baclofen would reduce between-category differences were that distinction not relevant. Thus, while it is safe to conclude that baclofen changed the structure of representations in PPC, it is more difficult to conclude that baclofen selectively alters within-category distances. This caveat may warrant greater emphasis in the Discussion.

11) I found Figure 5 and Figure 6 to be overkill. I think a basic schematic showing what within vs. between category distances refer to would be useful, but I don't think it is necessary to show schematics of how the categorization index might change in both cases (i.e., for PPC and then for OFC/pHIP). As I note above, I think the categorization index could be skipped altogether and the authors could go straight to the actual data separated by each condition/group/session.

12) In the subsection “Baclofen disrupts within-category odor discrimination in PPC”: “[…] associative connections in PPC are involved in preserving representational differences among odors belonging to the same category." Is it possible, instead, that baclofen 'blocks' learning of within category differences? Does the behavioral evidence indicate whether within-category discriminations improved from the time of fMRI session 1 to fMRI session 2?

13) A figure showing the behavioral discrimination performance in the pre and post- tests, split by condition, might be useful.

---

## [Author Response]

Essential revisions:

1) The pattern of data needs further explanation and unpacking. For instance, in Figure 5 – in support of their conclusion that baclofen leads to a reduction in pattern separation for within category odors (and not across category odors), they show changes in pattern distance for both placebo and drug. Is it possible that the effects noted are driven by greater pattern separation in controls over time? (which I think is what is being graphed)? This would lead one to a different interpretation about the results – that somehow after initial exposure without drug, the brain continues to differentiate – more of a consolidation effect? I may be confused however – but to help the reader better understand the pattern of results, it would be important for all the data to be plotted, the actual pattern distance Pre and Post separately instead of just the difference scores.

We thank the reviewer for this insightful observation. Admittedly the effects on within-category pattern separation (original Figure 5) were indeed driven more by changes in the placebo group over time, as opposed to those in the baclofen group. We welcome the reviewers’ alternative interpretation of these findings, and believe that it provides a more plausible explanation of the session effects in the data. Of note, this alternative conceptualization – greater pattern separation over time in the control subjects – is in close accordance with an earlier olfactory perceptual learning study from our lab, where prolonged passive exposure to one target odor increased its discriminability from categorically related odors (Li et al., Neuron, 2006). Viewed in this context, it can be reasonably inferred that baclofen interferes with olfactory pattern separation in PPC, possibly reflecting a disruption in consolidation.

As to the comment about plotting all of the data, rather than pre-post subtractions of pattern distances, we agree that this is a good idea. In revised Figure 5 (and also Figure 6 and Figure 7), we have reorganized the data presentation to include separate bar plots for each of the eight conditions (placebo within and across distances, pre and post; baclofen within and across distances, pre and post). We have also modified the Results subsections “Baclofen interferes with within-category pattern separation in PPC”, second paragraph and “Baclofen disrupts category coding in OFC and impedes within-category generalization in pHIP”, last paragraph and Discussion (first, second, sixth and last paragraphs) correspondingly.

2) Also, since there was not a significant main effect on any measure of odor perception or discrimination – indeed it looks as if everyone is doing better in the pre- versus the post assessment – how is this reconciled with the relationship between piriform odor representations and behavior?

The reviewer brings up a relevant question about the relationship between odor representations in the brain and corresponding behavior. Table 1 shows that there was no significant interaction between group (placebo/baclofen) and session (pre/post) on perceptual measures of odor identification, discrimination, or category descriptor ratings, nor any differences in odor pairwise similarity ratings, as a proxy measure for categorical perception.

There are at least three possibilities to account for this discrepancy. One possibility is that our behavioral tests were simply not sensitive enough (and less sensitive than the fMRI pattern measures) to detect changes in perceptual performance. Across both placebo and baclofen groups, there was a general trend towards improved performance, likely reflecting effects of training and exposure (e.g., thresholds and similarity ratings). As such, any further subtle effect of baclofen on perception may not have emerged beyond these training effects per se. Related to this, even if the three-way forced-choice pinene triangle test was arguably the most “sensitive” or difficult test of odor discrimination, this test might not have revealed a significant change if the perceptual learning effects had been confined to those odors presented repeatedly during the fMRI experiment (as opposed to the pinene stimuli which were encountered much less frequently).

A second possibility is that because all of the subjects were explicitly informed of the categorical features of the odor stimuli, there may have been an implicit tendency to anchor their perceptual responses to semantic categorical attributes. Moreover, even at baseline, all of the odors were easy to discriminate and highly familiar, and subjects were regularly called upon to make category judgments of the odors throughout the experiment. Thus, even though there was reorganization of categorical representations in PPC, these factors could have obscured our ability to observe perceptual plasticity across the set of odors, either for the placebo group or the baclofen group, despite significant group-by-session interactions in PPC.

Finally, one reasonable explanation for the brain-behavior discrepancy is that while changes in piriform odor representations (between placebo and baclofen groups) might have induced parallel changes in perception, it is possible that other brain areas would be able to compensate for these perturbations, helping to stabilize olfactory perceptual performance. For example, in the placebo group, increased within-category pattern separation (from pre to post) in PPC (Figure 5) would be counteracted by decreased within-category pattern separation in pHIP (Figure 6), resulting in no detectable change at the behavioral level. We have modified the in the seventh paragraph of the Discussion section to include these discussions.

Insofar as the placebo subjects exhibited greater within-category pattern separation in PPC (as discussed in Comment #1 above), we also tested session effects on perceptual behaviors within each group separately. Across-session improvements in detection thresholds and similarity ratings were significant in the placebo group (threshold: F_1,17_ = 6.11, P = 0.024; similarity: F_1,17_ = 4.93, P = 0.040), but not in the baclofen group (threshold: F_1,13_ = 0.60, P = 0.45; similarity: F_1,13_ = 1.24, P = 0.29). These findings suggest that there was some effect of odor exposure on olfactory perception in placebo subjects, coinciding with their fMRI pattern changes. However, because the differences between groups were not large enough to reveal a session × group interaction, we have chosen not to include these observations in the revised manuscript. If the editors and reviewers think that this information is useful to include, we are happy to do so.

3) I am a little concerned about the fact that differences are only seen for the fine-grain discriminations (within-category). Since the drug did produce significant sleepiness, is it possible that the tiredness and sedation associated with the drug may have also been associated with a lack of attention to the hardest discriminations? Their results still show that piriform cortex is not conducting this discrimination well but that would be an entirely different spin on the results.

Beyond that, I think the results are interesting. Since differences are seen across areas in the pattern of changes with Baclofen, it is easy to rule out any general effects of drug.

We thank the reviewer for this valid point, and agree that one strength of the findings is that different effects are seen across different areas and in the form of significant interactions, ruling out general drug effects on the brain. To investigate the possibility that baclofen-related sleepiness is associated with worse performance on the more difficult tests of discrimination, we tested the correlation between pre-post changes in sleep scale ratings and changes in fine odor discrimination performance (pinene triangle test) across baclofen subjects. This relationship was not significant (Spearman r = -0.33, P = 0.25, n = 14). Likewise, there was no significant correlation between sleep scale changes and within-category pattern distance changes in PPC across baclofen subjects (Spearman r = 0.11, P = 0.70, n = 14). We have added this information to the revised text in the last paragraph of the subsection “Baclofen interferes with within-category pattern separation in PPC “to indicate that there was no direct evidence that within-category odor discrimination in piriform cortex was related to level of sleepiness, on a subject-by-subject basis.

4) I think there are several confounds that are important to address. One is simply an order effect. Does the analysis take into account effects of prior odor? For example, one might expect that the MVPA would be very different to C1 after C2 than after W1 for example. This sort of short term "learning" or familiarity effect may be highly dependent on local synaptic interactions and thus susceptible to pharmacological manipulation. I would like to know that this ordering effect is either not present (or controlled for) and more importantly that the drug effect is independent of any ordering effect. Or not. If there is such an interaction, there is nothing wrong with that, but it would be an important factor in the interpretation.

This is a very interesting point that we had not considered. The six odors were presented in “pseudorandom” order such that each of the 6 odors was presented once in every 6 trials, and the order of these 6 odors was fully randomized within each set of 6 trials. Examination of our data indicates that across subjects and pre/post sessions, there were 39.3 ± 0.6 (mean ± SE) category repetition trials as opposed to 121.5 ± 0.9 category non-repetition trials. That ~75% of all trials were non-repetition trials suggests that category repetitions across trials did not have a pronounced effect on the findings. In any event, to examine whether category repetitions might introduce short-term perceptual biases of familiarity, working memory, or adaptation, complicating interpretation of the fMRI pattern findings, we conducted two new analyses.

First, we re-examined the original fMRI pattern dataset and MVPA model design. Here we took advantage of the fact that each of the scanning runs (28 trials per run) contained different proportions of odor category repetitions and non-repetitions. Our prediction was that runs containing greater numbers of category repetition trials (range across runs, 0%-52.2% repetition trials) would systematically bias pattern categorization effects in PPC. In this analysis, performed in each subject, the percent of category repetition trials was calculated for each odor in each run, and then these values were regressed against the corresponding measure of PPC categorization strength (indexed as across-category minus within-category pattern distances) for each run. The resulting correlation values from each subject were then submitted to one-sample t-tests or repeated-measures ANOVA across subjects. In the pre-drug session, the correlation coefficient across all subjects did not significantly differ from zero (r = -0.015 ± 0.024, t_31_ = -0.62, P = 0.54). There was also no difference between groups in the pre-drug session (one-way ANOVA, F_1,30_ = 0.097, P = 0.76), or in the interaction between session (pre/post) and group (mixed two-way ANOVA, F_1,30_ = 0.58, P = 0.45). These findings suggest that within-run effects of stimulus order did not affect the findings reported in PPC.

Second, we conducted a new MVPA analysis in which single-trial odor patterns were extracted from the “raw” fMRI responses in PPC. This procedure enabled us to directly separate the odor trials into category repetition and non-repetition conditions, and offered a way to test whether patterns evoked by a given odor under the repetition condition were significantly different from the non-repetition condition. Data pre-processing included extraction of fMRI signal intensity from each voxel in PPC for each scanning run, temporal de-trending with a high pass filter (cut-off frequency at 0.01Hz), and assembly into linear vectors of trial-specific voxel activity. Trial-by-trial pattern vectors were then sorted into “repetition” and “non-repetition” conditions for each of the 6 odors. For example, C1 trials preceded by a same-category trial (either C1 or C2) were assigned as repetition (rep) conditions; C1 trials preceded by a different-category trial (either M1, M2, W1, or W2) were assigned as non-repetition (non-rep) conditions. This procedure was independently applied to each of the 6 odors in turn. Then, for a given odor (e.g., C1_rep_), pair-wise correlation coefficients were calculated (1) between all pairwise repetition conditions (e.g., C1_rep_ versus C1_rep_), and (2) between repetition and non-repetition conditions (e.g., C1_rep_ versus C1_non-rep_). All possible pair-wise correlations were calculated, and then averaged across odors and comparisons. The main prediction was that if trial repetition had an impact on pattern category coding in PPC, then the correlation in (1) would significantly differ from the correlation in (2).

Group-wise statistical analysis indicated that there was no difference between the two types of correlations in PPC in the pre-drug session (paired-sample t-test, t_31_ = 0.42, P = 0.68), nor did a mixed three-way ANOVA reveal any difference in the interaction of order condition (within/between) × session × group (two within-subject factors of order condition and session; one between-subject factor of group; F_1,30_ = 0.51, P = 0.48). Together these new analyses suggest that sequence order of the odors does not have an impact on categorical pattern coding in PPC, either at baseline or in the context of drug. We have included these new analyses in the Methods subsection “Analysis of trial-wise order effects on odor patterns in PPC” and in the fourth paragraph of the revised Discussion.

5) A second confound is the novelty/familiarity across scans and the effect of drug. If I understand the design correctly, the subjects saw the odors for the first time pre-drug, then for the second time, post-drug. So the odors are novel and then familiar, if they are the same odors. Even if new odors are used the second time around, then the categories are still somewhat novel and then familiar. Either way, novelty is potentially confounded with drug effect. If the odors were not pre-exposed somehow in a serious way on a prior day (i.e. not same day as scanned), then I think this is important to note. This also brings into play effects of drug on consolidation of the initial exposure experience. I actually first thought of this when looking at the data on hippocampus. There is an effect of drug there, but really it looks like it is due to a loss of the normal decline in within-category pattern separation in the hippocampal BOLD response no? It seems to me that this may reflect a hippocampal generalization, essentially bringing closer together odors that are normally spaced far apart (i.e. diff categories) by virtue of their co-occurrence in the scanner/testing situation. Anyway again some consideration of this may be in order.

We are pleased that this point about novelty/familiarity was brought up. Although not discussed in detail in the original manuscript, we did in fact conduct a screening session on a different day prior to the main experiment. Screening was conducted to ensure that the subjects had normal olfactory abilities and were able to categorize the six odors reliably. During the screening session, subjects rated intensity, pleasantness, and familiarity for the six odors. Each odor was presented three times. Familiarity was rated on a visual analog scale with end-points of “extremely unfamiliar” and “extremely familiar”, and the cursor was reset at the mid-point on every trial. For analysis, the VAS ratings were scaled to a range from 0 (extremely unfamiliar) to 10 (extremely familiar). The mean familiarity rating of the six odors across all subjects was 7.68 ± 0.20, suggesting that subjects found the odor set to be relatively familiar. There was also no difference in odor familiarity ratings between subjects assigned to the placebo and baclofen groups (two-sample t-test, t_30_ = -1.18, P = 0.25).

Additionally, during the screening session, and following the odor ratings, subjects were asked to smell the six odors from glass bottles and to sort them into three categories. All subjects enrolled in the study were able to sort the odors appropriately into the three categories. We have added this information to the Methods subsection “Study design” to establish that subjects had been pre-exposed to the odors, found them to be familiar, and could associate them with categorical knowledge.

For the sake of clarity, the six odors used in the experiment were the same in the pre-drug and post-drug conditions.

With regard to the comment about hippocampal generalization, we think that this is an interesting idea. Indeed, in the placebo group there was a trend decrease (P = 0.097) of within-category pattern separation in pHIP, implying a tendency toward generalization for stimuli belonging to the *same* category. On the contrary, pattern generalization was not observed between odors of *different* categories. As such, the pattern changes in pHIP are not likely to be caused simply by virtue of their co-occurrence in the scanner/testing situation; if this had been the case, one might have expected to observe changes for both within- and across-category comparisons. We have now modified discussion of the hippocampal findings (Discussion, sixth paragraph).

6) I think just changes with time both within and across sessions are not being considered properly. Thus you might have changes in pattern from one odor to the next in a session or between the sessions (even independent of novelty). Eliminating this factor might give you a much clearer picture of what is happening it seems to me. And the authors have the perfect data to address this in the form of "same odor" contrasts. That is, if they analyzed the patterns evoked by the same odor from trial to trial and then took that as a baseline, they should control for such drift. Additionally, the classifier would perform a bit better at this than at within category comparisons. This would validate that there actually is a price to be paid for changing the odor within category. And again, it would be worthwhile to see that there was no effect of drug on this measure. Or to identify if there is an effect, since again this would impact the interpretation I think.

We thank the reviewer for the thoughtful suggestion of “same odor” contrasts. Based on this comment, it is clear to us that we did not adequately describe all of the steps involved in the correlation analysis in the original version of the manuscript. In fact, we did precisely what was suggested here, namely, we computed pattern distances between the same odor (using stimuli from different runs of course), and then subtracted these from the within-category and across-category pattern distances to control for within-session and between-session variations that could arise from training, familiarity, increasing boredom, drug effect, or even scanner drift or other artifact. In the revised manuscript we have expanded on our description of these methods to bring more attention to the utility of this baseline control contrast (subsections “Baclofen interferes with within-category pattern separation in PPC“, first paragraph and “Multivariate pattern analysis”, last paragraph).

7) I think some comparisons should be done by ANOVA or similar non-parametric multifactorial comparison. I noted one place in my review – the behavioral data – but I think possibly some of the other contrasts could be done this way, comparing group effects X within/across category for example.

We thank the editors and reviewers for these suggestions. While we had been using mixed-model ANOVA to test for drug effects on both behavior and fMRI pattern measures, it is true that we had used post-hoc two-sample t-tests for comparison between the two groups at the *pre-drug* session, to verify that there was no baseline difference between subjects who were to be randomized to the placebo or baclofen group. We had also used post-hoc paired-sample t-tests for comparisons between pre- and post-drug session within each group. In line with the reviewer’s comments, in the revised manuscript we now use mixed-model ANOVAs for all of the comparisons, allowing us to assess main and simple effects and their interactions within a single statistical framework. Please note that the resulting statistical effects are essentially unchanged from the originally reported findings.

8) Some of the critical data feel hidden. For example, at the end of the first paragraph of the subsection “Baclofen disrupts within-category odor discrimination in PPC”: the three-way interaction between session (pre/post), category (within/between) and drug (placebo/baclofen) is obviously a critical test but the authors don't fully lay out the data for inspection. That is, none of the figures show the full set of 8 data points (e.g., bars) in a single figure. Rather, the effect is unpacked via other analyses that involve some subtraction of conditions from one another. In particular, I think the "categorization index" might be eliminated altogether (e.g., Figure 5). Even for 5C (which uses a pre-post subtraction), I think it would be better to start with a figure that shows all of the data (i.e., the 8 bars) and then show the subtracted data. Showing the full set of data would also allow the reader to actually see the relative size of within category and between category distance effects (this is currently 'hidden' because all measures that are shown are pre minus post).

This comment echoes that in comment #1 above. We fully agree that the original presentation of the data needlessly obscured the response profiles and made things difficult to decipher. We have now fully modified all of the figures pertaining to the categorical pattern findings (new Figure 5, Figure 6, and Figure 7), and have eliminated the “categorization index” measure. The Results and Discussion sections have been updated accordingly (see subsections “Baclofen interferes with within-category pattern separation in PPC”, second paragraph and “Baclofen disrupts category coding in OFC and impedes within-category generalization in pHIP”, last paragraph and Discussion (first, second, sixth and last paragraphs).

9) In the last paragraph of the Results: “[…] while baclofen enhanced odor category coding in PPC […] “I believe the authors are referring to the fact that the change in "categorization index" was higher (more positive) in the baclofen group than the placebo group. But, as I note above, I find the categorization index to be a bit confusing and potentially misleading. It reflects the fact that, in PPC, the distance of within-category exemplars decreased relative to across-category exemplars. But I don't see how this is equivalent to "enhanced odor category coding." As shown in Figure 5, between-category distances did not change at all as a function of drug condition. This comes up again in the Discussion: "the net effect of baclofen was to enhance overall categorization in PPC […] “but that conclusion does not obviously follow from the findings.

We originally used “categorization index” for the purpose of keeping track of the direction of change in categorical structure of the group of odors, which fundamentally depends on the *relative* change of within-category and across-category distances. But we recognize that our presentation of the data in this manner made it confusing and difficult to understand the results. This issue was also raised in comments #1 and #8. Therefore, in our revised manuscript we have eliminated the categorization index measurement, and modified the Results, Discussion, and figures, which depict all 8 conditions (without any subtractions) and should simplify interpretation of the data.

10) One limitation is that we don't know how the task demands influenced the results. As the authors note in the Discussion (third paragraph), the fact that subjects were tasked with making category-level decisions, but not within-category discriminations, during fMRI scanning may well have been a factor in the observed results. This limitation is not necessarily a critical problem, but it does raise the possibility that baclofen would reduce between-category differences were that distinction not relevant. Thus, while it is safe to conclude that baclofen changed the structure of representations in PPC, it is more difficult to conclude that baclofen selectively alters within-category distances. This caveat may warrant greater emphasis in the Discussion.

We acknowledge that task-dependent factors could have influenced the impact of baclofen on across-category changes in PPC pattern representations. As noted in this comment, it is possible that the task focus on category-level decisions could have favored representation stability between categories. Interestingly, recent olfactory work by Chapuis and Wilson (Nat. Neurosci. 2011) suggests that task demands have a strong effect on whether rats use pattern separation or pattern completion strategies. Thus, it is fair to conclude that while we have no evidence to support baclofen-induced disruption of between-category codes, the nature of our task leaves open the possibility that in a different task where categorical distinctions were less relevant, baclofen might then have a modulatory influence on between-category differences. In the revised Discussion we have toned down our claims about within-category selectivity, and included some of the caveats and alternative interpretations (third paragraph).

11) I found Figure 5 and Figure 6 to be overkill. I think a basic schematic showing what within vs. between category distances refer to would be useful, but I don't think it is necessary to show schematics of how the categorization index might change in both cases (i.e., for PPC and then for OFC/pHIP). As I note above, I think the categorization index could be skipped altogether and the authors could go straight to the actual data separated by each condition/group/session.

As noted in some of our earlier responses, we have removed the categorization index as a measure of categorical change. Regarding original Figure 5 and Figure 6, we have condensed these into a single figure panel (new Figure 5), and have modified the schematic to be more compact and easier to grasp (we hope). Given the general complexity of all possible changes that could occur within and between categories, and with different implications for pattern separation and completion (generalization), we feel that including a basic schematic would be useful to help guide the reader. That said, if the reviewers and editors prefer to eliminate this figure, we would accept their decision.

12) In the subsection “Baclofen disrupts within-category odor discrimination in PPC”: “[…] associative connections in PPC are involved in preserving representational differences among odors belonging to the same category." Is it possible, instead, that baclofen 'blocks' learning of within category differences? Does the behavioral evidence indicate whether within-category discriminations improved from the time of fMRI session 1 to fMRI session 2?

This comment is similar to that brought up in comment #1. It does appear that the group difference in PPC was driven more by change in the placebo group than in the baclofen group; therefore, a “blocking” interpretation, in which baclofen prevents the natural tendency toward greater differentiation of within-category odors, might be a more plausible interpretation. As highlighted in our response to the previous comment, we have updated the interpretation of the PPC effect in the Results and Discussion sections.

In response to the question about behavioral evidence, we found that the similarity ratings of within-category odors (arguably the most relevant measure for assessing within-category discriminations) did not show evidence of improvement over time (main effect of session: F_1,30_ = 2.95, P = 0.15), nor was there a session × group interaction (F_1,30_ = 0.22, P = 0.64). Moreover, in the fine-odor discrimination task, which used different odors from those in the main experiment, no improvement was seen from pre- to post-drug session (main effect of session: F_1,30_ = 1.41, P = 0.24). Please see our response to comment #2 for discussion about possible explanations for the discrepancy between the fMRI pattern changes and behavior.

13) A figure showing the behavioral discrimination performance in the pre and post- tests, split by condition, might be useful.

We thank the reviewer for the suggestion. We have now included the behavioral discrimination plot in Figure 3.